# Elucidating variations in outcomes among older end-stage renal disease patients on hemodialysis in Fukuoka Prefecture, Japan

Aziz Jamal[1,2]*, Akira Babazono[1], Yunfei Li[1], Takako Fujita[1,3], Shinichiro Yoshida[1], Sung A. Kim[1]

1 Department of Health Care Administration & Management, Graduate School of Medical Sciences, Kyushu University, Fukuoka, Japan, 2 Health Administration Program, Faculty of Business & Management, Universiti Teknologi MARA, Selangor, Malaysia, 3 Department of Health Sciences, Faculty of Medical Sciences, Kyushu University, Fukuoka, Japan

* azizabduls.jamal@gmail.com

**Data Availability Statement:** The data used in our study cannot be publicly shared due to the restriction imposed by the Ethics Committee of Kyushu University. However, researchers who

## Abstract

Variations in health care outcomes and services potentially indicate resource allocation inefficiency. Therefore, this study was conducted to examine variations in mortality and hospitalization cases among end-stage renal disease (ESRD) patients receiving hemodialysis (HD) care from medical facilities located in 13 secondary medical care areas (SMAs) of Fukuoka prefecture, Japan. The research was designed as a retrospective, cross-sectional study using insurance claims data. The subjects of the study were older patients (over 65 years old) insured by the Fukuoka prefecture's Latter-Stage Elderly Healthcare Insurance. Using an electronic claims database, we identified patients with chronic kidney disease (CKD) who had received HD care from April 1, 2017 to March 31, 2018. The CKD status was identified using International Classification of Disease, 10th revision code, and HD maintenance status was ascertained using specific insurance procedure codes. A total of 5,243 patients met our inclusion criteria and their records were subsequently reviewed. About 73% (n = 3,809) of patients had admission records during the period studied. Thus, the data regarding hospital length of stay (LOS) and admission costs were analyzed separately. Significant differences in terms of increased risks in hospitalization were evident in a number of SMAs. An increase in mortality risk due to heart failure and malignancy was observed in two separate SMAs. Also, analyzed LOS, total hospitalization cost, and cost per day according to SMAs showed statistically significant variations. The findings highlight the magnitude of the burden of CKD and ESRD in the community. The high prevalence of ESRD, associated mortality, and hospitalized HD patients signal the need for clinicians to assume broader roles in measures against chronic kidney disease through involvement in community awareness programs. To improve patient outcomes, improvement of regional health care provision, the level of medical care, and the development of existing human resources are needed.

meet the criteria for access to this confidential data may request data access by emailing the administrative office of Bioethics Section (Medical Sciences), Academic Research Cooperation Division of Kyushu University at ijkseimei@jimu@kyushu-u.ac.jp.

**Funding:** The authors received no specific funding for this work.

**Competing interests:** The authors have no conflict of interest to disclose.

## Introduction

Japan has been witnessing a steady increase in the number of patients with end-stage renal disease (ESRD). The country recorded 339,841 ESRD cases in 2018, with a prevalence rate of 2,688 cases per million population [1]. With these numbers, Japan had the second-highest number of patients undergoing hemodialysis treatment in the world, after the United States. In comparison, in December 2017, the United States reported 746, 577 cases of ESRD, which translates to a crude prevalence of 2,204 ESRD cases per million population [2, 3]. Additionally, a report published by the Global Burden of Diseases, Injuries, and Risk Factors Study 2017 estimated that total of 697.5 million chronic kidney (CKD) cases being reported worldwide, with an estimated global prevalence as 9.5% of the population in 2017. The global prevalence for ESRD, on the other hand, was only 0.041% [4]. Given that the world population has reached 7.60 billion in 2017, this statistic translates to a 3.1 million ESRD patients dependent on dialysis care.

Survey data provided by the Japanese Society for Dialysis Therapy (JSDT) revealed an approximately 3% increase in the number of treated ESRD patients every year [1]. With a recorded 40,486 new dialysis patients in 2018, an increase in demand for dialysis care services is unavoidable. A total of 4,458 care facilities were available throughout Japan in 2018, indicating an additional 145 new facilities as compared to the previous year.

Providing dialysis care is necessarily expensive. Nevertheless, the government heavily subsidizes this cost, so the burden on patients is reduced. An older person with Latter Stage Elderly Healthcare Insurance, for example, pays only ¥10,000 (or ¥20,000 if the income level is categorized as high) per month for outpatient or inpatient services for each facility. Under the Services and Supports for Persons with Disabilities Act, the Japanese health care recognizes ESRD, which requires constant dialysis care, as a disability [5]. Therefore, the amount of money to be paid will be further discounted, making dialysis treatments free in most cases.

It is estimated that Japan spends ¥5 million to provide dialysis care per person each year [5]. Undoubtedly, the cost of providing such care incurs undue financial and administrative burdens on the government. The number of ERSD patients has now reached 339,841, equal to a total annual cost of ¥1.6 trillion (USD 14.9 billion). This indicates the medical cost for ESRD alone represents approximately 4% of the total health care budget of Japan.

The number of ESRD patients that have been treated in Japan varied statistically according to the Prefecture. Of the 47 prefectures in Japan, the ten prefectures with the highest number ESRD patients are *Tokyo* (32,682 patients), *Osaka* (24,070 patients), *Kanagawa* (21,664 patients), *Aichi* (18,783 patients), *Saitama* (18,541 patients), *Hokkaido* (16,060 patients), *Chiba* (15,525 patients), *Fukuoka* (15,137 patients), *Hyogo* (14,390 patients), and *Shizuoka* (11,158 patients). The ten prefectures with the highest rates per capita are (per million residents) *Tokushima* (3819.3), *Kumamoto* (3,758.7), *Miyazaki* (3,652.2), *Kochi* (3,546.7), *Oita* (3,546.3), *Kagoshima* (3429.4), *Tochigi* (3,329.9), *Wakayama* (3,224.6), *Saga* (3,139.2), and *Gunma* (3103.5) [1]. Early attempts to examine the statistical variations in ESRD patients across Japan can be found in Usami et al works [6–8]. Comparing the annual number of patients with ESRD who had initiated dialysis maintenance from 1982 to 1998 across 11 regions in Japan, a significantly higher annual ESRD incidence was observed in the *Kyushu*, *Shikoku*, and *Okinawa* regions. The slope of regression lines presented by the authors also indicated an increasing rate of ESRD prevalence in the *Kyushu*, *Sapporo*, and *Okinawa* regions over a period of 17 years [6–8].

In Japan, about 60% of ESRD patients are currently treated with hemodialysis (HD) [1]. Regardless of the type of treatment received, approximately 30,000 ESRD patients die each year from various complications. With an annual crude mortality rate between 9.6% and 10%

in recent years, heart failure, infectious disease, malignancy, and cerebral stroke have consistently been named the most common factors leading to ESRD patient death [1]. In comparison, in December 2018 alone, 7,301 patients died of heart failure, 6,640 died of infection, 2,609 died of malignancy, and 1,859 died of cerebrovascular diseases.

The Japanese government has recognized variations in terms of ESRD prevalence, disease pattern, and patient survival outcomes at the national and the prefectural levels [6–10]. The annual collection of JSDT data, for example, provides vital statistical comparisons of ESRD population and the outcomes between and among prefectures. As the increasing number of patients with ESRD reflects a significant public health concern, the government—through the Kidney Disease Control Commission, had developed a number of strategies to promote measures against kidney disease [11]. These efforts include the improvement of regional health care provisions and the level of medical care.

Variations in healthcare are not necessarily unwarranted. In many circumstances, health and outcomes variations are simply the results of random differences in population and demographic characteristics. However, variations that negatively influence health outcomes are necessarily unwanted as they might signal the presence of social inequity, service quality issues, and resource allocation inefficiency [12, 13]. It is currently acknowledged there are three common causes of these unwanted variations in health care. Variations in the health outcomes might be attributed to the variations in the use of effective care due to differences in clinical knowledge, differential rates of diffusion and adoption of innovation [14, 15]. At health care level, unwanted variations may be related to sub-optimal productivity and health delivery inefficiency, which are greatly determined by factors such as clinician characteristics, service capacity, areas of care, and the effective management of care facilities [15, 16]. Additionally, variations could also originate from the differences of service utilization patterns such as over- and under-utilization of specific medical services. This type of variation is largely influenced by the supply and demand of care services, which are driven by health care access, cost, economic incentives, and the provision of healthcare payment system [13, 15]. In the context of HD care and health care delivery among patients with ESRD, variations in patient outcomes depend heavily on a number of factors, including clinical practice pattern, patient's health status and comorbidities, and the quality of HD care and service delivery [10, 17]. An increase in utilization rate of health care services among patients with HD, in addition to HD care itself, is also anticipated due to poor health and increased number of comorbidities [18]. It is also expected that these patients would more frequently use hospital admission services to address multiple health issues, that could lead to a significant increase in overall health care costs.

Past studies have placed little emphasis on examining the sources of variations, as they mostly analyzed the incidents and prevalence of ESRD itself. Statistical comparisons were also performed using data aggregated at the national level, and little information on health care service utilization among patients with ESRD was provided. As the health care in Japan is managed by the prefectural government, which organizes its delivery according to secondary medical care areas (SMAs), it is therefore necessary to examine the extent to which the outcomes and health care variations exist at prefectural level. These findings should provide a better understanding of the magnitude of these variations, the contribution of these variations to the overall differences in health outcomes across nation, as well as offering meaningful data for improvement of regional health care provisions and the level of medical care.

This study therefore aims to examine the variations in mortality outcomes of HD patients receiving care in Fukuoka prefecture, based on reported all-cause mortality cases and the mortality associated with specific diagnoses i.e., heart failure, infectious disease, malignancy, and cerebral stroke. Additionally, this study aims to examine hospitalization rates based on the available data. It also aims to evaluate the utilization of health care services (i.e., length of stay

(LOS), total hospitalization costs, and costs per day) among HD patients. These findings might provide insight into the cause of regional variations in HD care services and delivery in Fukuoka prefecture to target the potential areas in need of improvement. Policymakers will also be better informed on how best to allocate health care spending.

## Methods

### Study design

This study was designed as a retrospective, cross-sectional study. Using the electronic database, all ESRD patients with active insurance status and that have received HD care between April 1, 2017 and, March 31, 2018, were identified.

### Study location

This study was conducted in Fukuoka prefecture, Japan. Located on the Kyushu island in the southern region of Japan, statistics in 2018 estimated that its population had reached 5,100,000 [19]. With the geographic area of 4,987 km$^2$, Fukuoka prefecture is currently ranked 9th among 47 prefectures in Japan, in terms of population size, and ranked 8th in terms of population density (with 1,024 people per km$^2$) [19]. There is also a considerable older population. A recent estimate indicates that 28% of population is over 65 years old. Currently, Fukuoka prefecture has 12 districts and 60 municipalities [19].

Health care delivery is organized according to 13 SMAs, which fall under the jurisdiction of the prefectural government. Until December 2017, there were 435 general hospitals, 4,666 medical clinics, 61 psychiatric hospitals, and 3,094 dental clinics available across SMAs [19]. Hemodialysis care is provided by a total of 198 facilities that located throughout the prefecture [1]. These facilities are either operated as an ambulatory hemodialysis unit attached to a hospital, or freestanding hemodialysis centers or clinics. As patients typically require frequent hemodialysis sessions (2–3 times per week), the selection of hemodialysis facility is fixed to a specific SMA located in or close to the patient's residential area. Patients, however, are allowed to change the assigned hemodialysis facility on specific occasions. For example, this change could be due to a change in residential address, or due to medical requirements that can only be provided by specialized hospitals in other SMA. **Fig 1** depicts the organization of health care in Fukuoka prefecture according to SMAs.

### Population (patient identification)

The population of this study was ESRD patients with active insurance status and that have received HD care between April 1, 2017 and, March 31, 2018. We used the International Classification of Disease, 10th revision, code N18.0, to identify chronic kidney disease diagnosis and specific insurance codes to confirm patients' HD maintenance status. Claims records provide information on HD facilities. Because it is possible for a patient to receive HD care from facilities located in several SMAs, we identified the SMA based on the location of the facility for which a patient most frequently receives HD care. If we were unable to determine a specific SMA due to the complexity of claims records, SMA based on the patient's residential address would otherwise be used.

**Inclusion & exclusion criteria.** We included all patients who had received HD care during the study period. This inclusion was limited to patients who received HD exclusively as the main treatment for ESRD. Thus, patients who primarily received peritoneal dialysis but also needed intermittent HD care were excluded. Hemodiafiltration (HDF) is considered a distinct treatment modality. Therefore, we did not include HDF patients in our study. We also

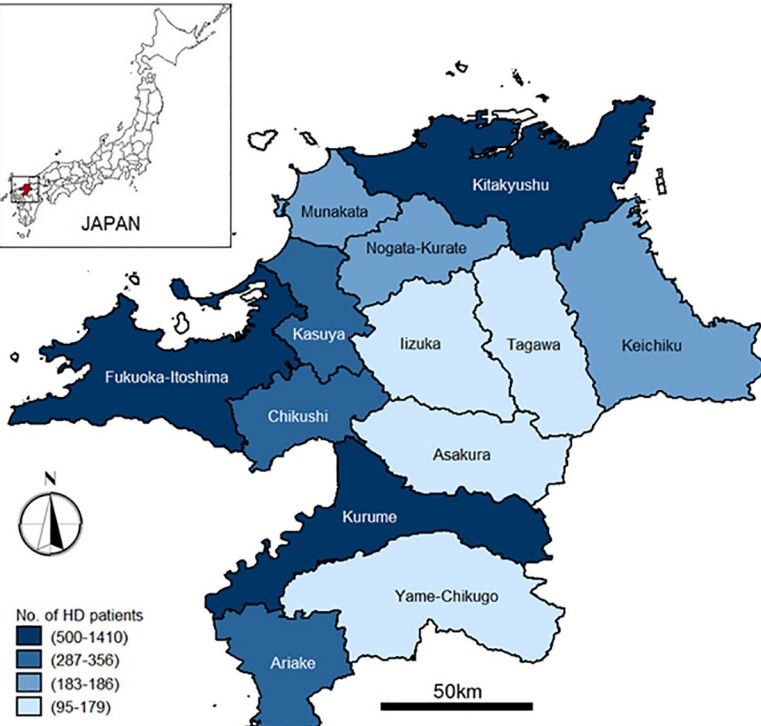

**Fig 1. Secondary Medical Care Areas (SMAs) of Fukuoka Prefecture.** The SMA is the unit of secondary care governed by a prefecture according to Japan Medical Service Law. The small inset shows the location of the prefecture in Japan. Reprinted from National Land Information Division, National Spatial Planning and Regional Policy Bureau, MLIT of Japan and maps were created using shapefiles based on Secondary Medical care Area (SMA), downloaded from the National Land Information Division Web Mapping System (Ministry of Land, Infrastructure, Transport and Tourism, Japan) at https://nlftp.mlit.go.jp/ksj/gml/datalist/KsjTmplt-A38.html, under a CC-BY license, with permission from the National Land Information Division, National Spatial Planning and Regional Policy Bureau, MLIT of Japan, original copyright ©2014.

excluded patients who were younger than 65 years old by April 1, 2017 (*n* = 224). As the insurance scheme is intended to provide coverage for those over 65 years old, we assumed the inclusion of younger patients in the database represents a coding error. We included patient who had received at least 24 HD sessions; thus, patients who had undergone fewer than 23 sessions were excluded (*n* = 625), as we could not rule out whether such short-term HD was provided to address issues related to ESRD or other conditions.

**Sample size.** A total of 5,243 patients were identified and found to meet the inclusion and exclusion criteria. Subsequently, the claims data of all 5,243 patients were retrospectively reviewed and the reported all-cause mortality, diagnosis-specific mortality, and hospitalization cases were analyzed according to 13 SMAs of Fukuoka prefecture. The data identified that 73% of HD patients (*n* = 3,809) had one or more hospital admission episodes during the period studied. To examine these patients' utilization of hospital admission service, separate analyses on the LOS, total hospitalization costs, and costs per day were performed according to SMAs. **Fig 2**. presents a diagram of patient identification process and records selection.

## Data source

Data were primarily obtained from insurance claims records submitted to the Fukuoka prefecture Latter-Stage Elderly Healthcare Insurance Association. In Japan, citizens aged ≥ 75 years old are entitled to receive medical protection from the Latter-Stage Elderly Healthcare

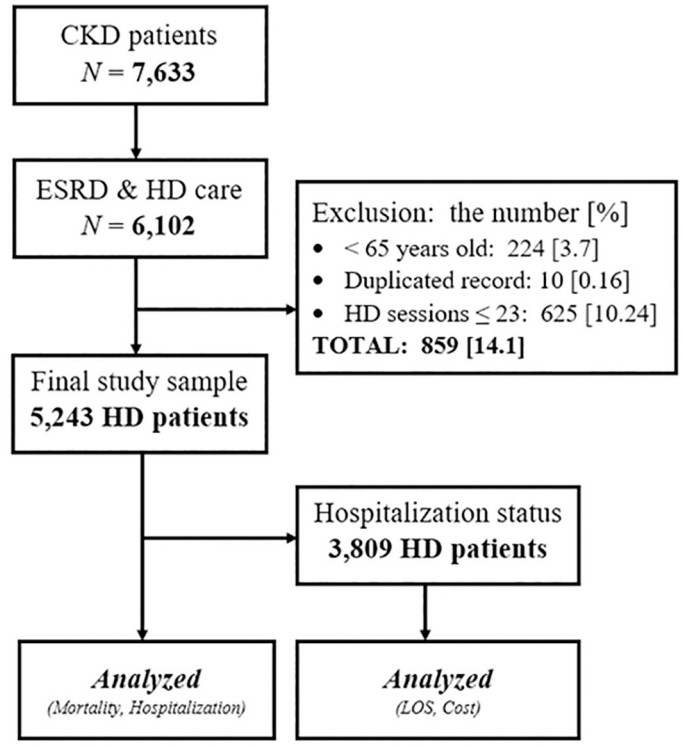

**Fig 2. Diagram of patient identification and record selection.** A total of 5,243 hemodialysis patients were identified and found to meet inclusion criteria. The claims records of each patient were retrospectively reviewed for one year (April 1, 2017–March 31, 2018).

Insurance Scheme. The insurance coverage extends to patients aged 65–74 years old with specific disabilities, e.g., end-stage renal disease (ESRD) requiring constant dialysis care. The insurance claim records are maintained electronically. Data were extracted from the insurance database using SQL Server 2014 program (Microsoft Corporation, Redmond, WA)

## Definition of variables

The claims data provides basic information on patient demographics and medical history. The patients were categorized by sex and two age categories: 65–74 years old, and $\geq$ 75 years old. The International Classification of Disease, 10th revision (ICD-10) codes, were used to identify the status of dementia (F00, F01, F02, F03, F04, F05.1, F06.6, F06.8, F09, F10.7, F11.7, F12.7, F13.7, F14.7, F15.7, F16.7, F17.7, F18.7, F19.7, G30, G31.1, G31.9, G32.8, G93.7, G94, R54). To assess the presence and the severity of comorbid conditions, the Charlson Comorbidity Index (CCI) was adapted and used [20]. This validated instrument assigns weights (1, 3, or 6) to each of 19 major comorbid conditions that likely influence treatment prognosis and survival outcomes. In this study, the weight was modified ($m$-CCI) to exclude chronic kidney disease. Subsequently, three categories were created based on the calculated weights to represent the degree of severity: mild ($m$-CCI $\leq$ 1), moderate ($m$-CCI = 2–6), and severe ($m$-CCI $\geq$ 7). The use of CCI to measure the severity of comorbid conditions based on the insurance claims data in Japan has been widely and reliably documented [21, 22].

Outcome variations have been analyzed and presented according to secondary medical care areas (SMAs). SMAs are small geographical areas into which each prefecture organizes its health and medical care delivery. All 47 prefectures in Japan are required by law to establish their own SMAs. Presently, Fukuoka prefecture has 13 SMAs, with *Fukuoka-Itoshima* and *Kitakyushu* constitute the two largest SMAs.

## Definition of outcomes

The primary outcomes included all-cause mortality, as well as mortality cases among patients with heart failure, infectious disease, malignancy, and cerebral stroke (i.e., diagnosis-specific mortality). The insurance claim code 202 was used to determine mortality status. In the insurance data, such a code signifies loss of insurance eligibility due to death. Several epidemiological studies in Japan have used a claims-based definition of death previously. Past studies validating such a mortality status indicate high specificity and positive predictive values, suggesting a low likelihood of outcome misclassification [22, 23]

As the records are primarily used for reimbursement purposes, the underlying causes of death are not usually recorded. Therefore, to associate specific diagnoses with mortality status, we characterized the main diagnosis reported at the last admission or health encounter as 'mortality-related'. If there were competing diagnoses (e.g., more than one chronic diseases reported as the main diagnoses at the last admission), records were further examined to identify a specific diagnosis that existed for the longest period, and for which the patient continued to receive medical care prior to death. Only deaths that occurred within 90 days after the onset of any infectious diseases were considered to be 'infection-related'. Diagnoses reported at the last admission or encounter other than heart failure, infectious disease, malignancy, and cerebral stroke were regarded as 'other specified diagnoses' and thus were not analyzed in this study. ICD-10 codes adapted from the ICD-10-CM codes used for United States Renal Data System (USRDS) reporting were used in this study to identify the main diagnoses. Both inpatient and outpatient claims records were reviewed and used to ascertain the diagnoses. For a specific diagnosis or comorbidity code to be considered valid and included in analysis, additional claims record specifying treatments, procedures, or prescriptions must present and indicative of specific conditions being treated. We did not, however, set a specific duration or a frequency for the condition must appear on the claims record as part of the inclusion criteria. Table 1 presents the ICD-10 codes (equivalent to first 4 characters of ICD-10CM) used to identify main diagnoses associated with mortality.

The hospitalization status of hemodialysis patients—defined by the presence of one or more inpatient records during the study period—was also considered as a primary outcome.

**Table 1. ICD-10 codes used to define the cause of hospitalization.**

| Main diagnosis | International Classification of Disease, 10th Revision (ICD-10) code |
|---|---|
| Heart failure | I11.0; I13.0; I42.0–I43; I50.0–I50.9 |
| Infectious disease | A00.0–A32.9; A35–A48.0; A48.2–B44.7; B44.8–B78.0; B78.7–B99<br>D86.0–D86.9; G02; G14; H32; I32; I39; J02.0; J03.0; J17; J20.0–J20.7; K90.8; L44.4; L94.6; M00.0–M00.8; M02.3; M60.0; N34.1; T802; T82.7 |
| Malignancy | C00.0–C43.9; C45.0–C75.9; C76.0–D03.9; D05.00–D09.9 |
| Cerebral stroke | I60.0–I66.9 |

Note: ICD-10-CM codes that sufficiently describe the condition at the at category level (the first 3 characters), or by the 4th character of subcategory level, were used. Codes that only distinguish the condition at the 5th, 6th or 7th character of subcategory level were dropped from the list.

The secondary outcome, on the other hand, refers to the length of hospital stay (LOS). For patients with hospital admission records, LOS was calculated by subtracting the day of admission for the day of discharge for every episode of hospitalized care per patient in one year. Additionally, the total hospitalization costs—defined as one-year cumulative costs paid by the insurance association for medical care requiring hospitalization—was also calculated. These total hospitalization costs included the costs of surgery, procedures, medication, and diagnostic tests provided during hospitalization [24]. We estimated the costs per day by dividing the total hospitalization costs with the cumulative days of inpatient stay (LOS).

## Data analyses

We performed the statistical analyses in two stages. The first stage was focused on the descriptive and inferential analyses of all-cause mortality, diagnosis-specific mortality, and hospitalization cases reported across Fukuoka prefecture's SMAs. These statistical analyses used the insurance claims data of 5, 243 HD patients included in the study. The second stage was aimed at examining the LOS, total hospitalization costs, and costs per day of HD patients who were hospitalized at least once during the study period. Our claims data identified a total of 3,809 patients who had received one or more medical care episodes requiring hospitalization. The remaining 1,434 patients without such hospitalization records were excluded from these analyses as we could not accurately calculate their LOS and hospitalization cost.

The descriptive statistics for the categorical variables in this study were expressed in numerical values and percentages. Mean (*M*) and standard deviation (*SD*) were used to present the data for continuous variables. The comparisons of all-cause mortality, hospitalization, and diagnosis-specific mortality cases according to sex, dementia status, and SMAs were based on Pearson's chi-squared and Fisher's exact tests. We considered age groups and *m*-CCI categories as ordered categorical variables. Therefore, the Mantel-Haenszel Chi-squared test to determine associated trends was used as recommended by several statisticians [25]. Univariate logistic regression analyses were conducted to examine the association between SMAs and all-cause mortality, hospitalization, and diagnosis-specific mortality. Sex, age categories, dementia status, *m*-CCI categories, were later included in the multiple logistic model, and regressed together with SMAs to examine possible associations with studied outcomes. The results of the univariate and multiple logistic regression analyses have been presented as an odds ratio (OR) with a 95% confidence interval (CI). The goodness-of-fit of the logistic models was determined via regression diagnostics using *Cragg & Uhler's (Nagelkerke)* pseudo-R-squared test. This test value is useful in examining the fitness and the complexity of logistic model, as it presents a normalized version of *R*-squared value computed from the likelihood ratio. Thus, the calculated pseudo-*R*-squared value has connection with the Wald statistics for overall association [26, 27]. The use of this statistic to examine 'the explained variance' in a fitted model has been widely adopted in the scientific literature.

Data regarding length of hospital stay (LOS), total hospitalization cost, and cost per day were analyzed using the Generalized Linear Model (GLM). This model provides superior estimates compared to a normal linear regression model due to its alternative approach in dealing with problems associated with skewed LOS and cost data distribution [28]. In this model, outcome variables were assumed to have a gamma distribution. For the model containing LOS as an outcome variable, the identity-link was used to assume an additive effect [29]. The fitted coefficients, therefore, represent the number of inpatient days a patient stayed in a hospital located in specific STMs. Log-link, on the other hand, was used in GLMs that contained total hospitalization cost and cost per day as outcome variables. Models incorporating a log-link assumed multiplicative or proportional effects [29], provide an estimate of the cost ratio

incurred by patients receiving hospitalized care in specific SMAs. The marginal means of LOS, total cost and cost per day were calculated to indicate the estimated value of inpatient days, total hospitalization cost, and cost per day. The cost-related statistics were originally calculated in Japanese Yen. For convenience, these values were converted to the US dollars to allow easy interpretation and comparison. By April 1, 2018, the exchange rate of US$ 1 was JP¥ 112. The goodness-of-fit of GLM was assessed based on *Cameron & Windmeijer's* R-squared test [30]. To the best of our knowledge, it is the only statistical test that provides information on the GLM model fitness based on calculated model variance.

To determine the presence of variations in LOS, total hospitalization costs, and costs per day according to the SMAs, one-way ANOVAs were conducted, together with post-hoc analyses using *Games & Howell*'s pairwise multiple comparisons, to determine the variability between specific combinations of SMAs. Box-Cox transformation was performed on data containing LOS, total hospitalization cost, and cost per day, to address issues related to skewed data distribution [31]. Distribution normality was subsequently inspected using a histogram, and statistically assessed using the *Anderson-Darling* normality distribution test. This normality distribution test was chosen because it allows the specification of exponential, lognormal, and gamma distributions, from which the critical values are calculated [32]—thus offering a greater test sensitivity. As equal variances were not assumed, *F* statistics based on Welch's ANOVA were reported. Welch ANOVA provides unbiased estimates, and its use has been recommended many statisticians when dealing with data with unequal variances in data [33, 34]. The adjusted omega-squared ($\omega^2$) values representing Welch's *F*-test effect size were calculated using formula (1) below:

$$\omega^2 = \frac{df_{bet}(F-1)}{df_{bet}(F-1) + N_T} \tag{1}$$

Where $df_{bet}$ represents the degree of freedom between groups, *F* represents the Welch *F*-statistics, and $N_T$ represents the population sample size [33, 34]. All reported *p*-values were 2-tailed, and the level of significance was set at $P < .05$. Stata Statistical Software: Release 14 (StataCorp LP, College Station, TX), and IBM SPSS Statistics for Windows, version 23.0 (IBM Corp, Armonk, NY), were used to analyze the data.

## Ethical considerations

This study used anonymized claims insurance data. Thus, the requirement to obtain informed consent was waived in accordance with the Ethical Guidelines for Medical and Health Research Involving Human Subjects in Japan [https://www.mhlw.go.jp/file/06-Seisakujouhou-10600000-Daijinkanboukouseikagakuka/0000080278.pdf]. This study was also approved by the Institutional Review Board of Kyushu University (Clinical Bioethics Committee of the Graduate School of Medical Sciences, Kyushu University).

## Results

### Patient characteristics

Data from 5,243 hemodialysis patients were included and analyzed. Descriptive statistics revealed a large number of patients were males ($n$ = 3,254, 62.1%), aged $\geq$ 75 years old ($n$ = 3,111, 59.3%), and living with multiple comorbidities categorized as severe ($n$ = 2,137, 40.8%). A total of 1,217 patients were living with dementia, constituting a 23.2% of studied population. *Fukuoka-Itoshima* had the most ($n$ = 1410, 26.9%), and *Asakura* had the fewest hemodialysis patients undergoing treatment ($n$ = 98, 1.8%). Additional analyses were

performed to compare sex and age categories according to *Fukuoka* Prefecture's 13 SMAs. Significant differences were observed in five SMAs: *Yame-Chikugo* ($\chi^2_{12}$ = 5.40, $p$ = .02), *Ariake* ($\chi^2_{12}$ = 5.30, $p$ = .02), *Iizuka* ($\chi^2_{12}$ = 5.97, $p$ = .02), *Nogata-Kurate* ($\chi^2_{12}$ = 5.83, $p$ = .02), and *Kitakyushu* ($\chi^2_{12}$ = 5.25, $p$ = .02).

## All-cause mortality and hospitalization cases

Within one year, a total of 614 patients died, translating to a 11.7% crude mortality rate. The median age of the patients that died was 80 years old ($M$ = 79.3, $SD$ = 7.6). A significant difference was observed in the age categories, with more patients in the age category of $\geq$ 75 years old ($\chi^2_1$ = 46.13, $p <$ .001). Similarly, all-cause mortality was higher among males ($\chi^2_1$ = 6.56, $p$ = .01), patients with the status of dementia ($\chi^2_1$ = 96.34, $p <$ .001), and patients living with moderate and severe comorbidities ($\chi^2_2$ = 38.56, $p <$ .001). On the contrary, no significant variations were observed in all-cause mortality when the statistics were compared according to SMAs.

A high number of patients receiving hospitalization care was recorded ($n$ = 3,809, 72.6%). The median age of hospitalized patients was 77 years old ($M$ = 77.2, SD = 7.2). Significant variations in age categories were also observed. In particular, a higher number of patients aged $\geq$ 75 years old receiving one or more care episodes were observed to require hospitalization compared to the younger age group ($\chi^2_1$ = 62.04, $p <$ .001). While there was no significant difference in the number of patients who received hospitalized care according to sex, statistically significant differences between patients, grouped by dementia status ($\chi^2_1$ = 92.22, $p <$ .001), $m$-CCI categories ($\chi^2_2$ = 153.24, $p <$ .001), as well as among patients who received care from 13 SMAs, were statistically evident ($\chi^2_{12}$ = 110.53, $p <$ .001). **Fig 3** shows the distribution of patients according to the SMAs, and the statistics regarding all-cause mortality and hospitalization cases during the period studied.

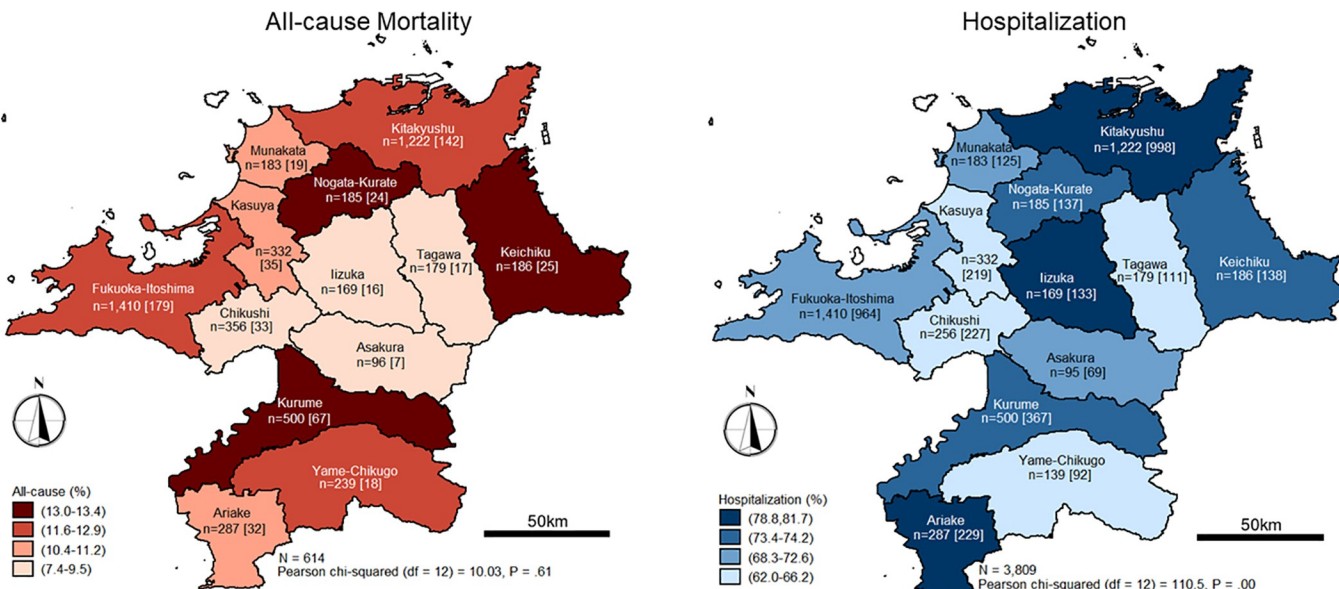

**Fig 3. Reported all-cause mortality and hospitalization cases according to SMAs (April 1, 2007–March 31, 2008).** Figures enclosed in square brackets represent cases. Reprinted from National Land Information Division, National Spatial Planning and Regional Policy Bureau, MLIT of Japan, and Maps were created using shapefiles based on Secondary Medical care Area (SMA), downloaded from the National Land Information Division Web Mapping System (Ministry of Land, Infrastructure, Transport and Tourism, Japan) at https://nlftp.mlit.go.jp/ksj/gml/datalist/KsjTmplt-A38.html, under a CC-BY license, with permission from National Land Information Division, National Spatial Planning and Regional Policy Bureau, MLIT of Japan, original copyright ©2014.

### Diagnosis-specific mortality

Reported mortality cases were analyzed, with selected primary diagnoses assigned at the last episode admission or health care encounter (i.e., heart failure, infection, malignancy, and cerebral stroke). For heart failure-associated mortality, no significant difference was observed in the age, sex, dementia status, or $m$-CCI categories, and cases reported in each SMA. However, a significant difference was observed in SMAs for infection-associated mortality between age groups and dementia status, although reported cases according to sex, $m$-CCI categories, and SMAs did not differ significantly. The mortality cases of patients with malignancy were statistically different when compared according to the $m$-CCI categories. However, mortality cases associated with cerebral stroke were not statistically different in terms of age group, sex, dementia status, $m$-CCI category, and SMA. **Table 2** summarizes the results of the descriptive statistics for all-cause mortality, hospitalization, and diagnosis-specific mortality according to the age group, sex, dementia status, and $m$-CCI categories. **Fig 4** illustrates the number of patients with specific diagnoses according to SMAs and the corresponding mortality cases (enclosed in square brackets), along with the results of Fisher's exact test.

### Outcomes association with SMA

To examine the association of measured outcomes (i.e., all-cause mortality, hospitalization, and diagnosis-specific mortality) with specific SMA, logistic regression analyses were performed. These analyses were performed in stages. Firstly, we developed univariate logistic regression models to examine the association of each SMA with measured outcomes. Secondly,

**Table 2. Summary of descriptive statistics examining all-cause mortality, hospitalization, and diagnosis-specific mortality according to age, sex, dementia status, and m-CCI categories indicating the severity of comorbid conditions.**

| | | | | | | | Diagnosis-specific mortality[a] | | | | | | | | | | | |
|---|---|---|---|---|---|---|---|---|---|---|---|---|---|---|---|---|---|---|
| | All-cause Mortality | | | Hospitalization | | | Heart Failure | | | Infection | | | Malignancy | | | Cerebral stroke | | |
| | *n* | (%) | *p* | *n* | (%) | *p* | died | (%) | *p* | died | (%) | *p* | died | % | *p* | died | % | *p* |
| **Age**[b] | $M = 79.3$, $SD = 7.6$ | | | $M = 77.2$, $SD = 7.2$ | | | $M = 79.3$, $SD = 6.9$ | | | $M = 78.5$, $SD = 7.1$ | | | $M = 78.3$, $SD = 7.0$ | | | $M = 79.7$, $SD = 8.3$ | | |
| 65–74 (%) | 172 | 8.1 | < .001 | 1424 | 66.8 | < .001 | 10 | 11.6 | .17 | 21 | 8.1 | .01 | 10 | 5.9 | .20 | 15 | 9.4 | .21 |
| ≥75 (%) | 442 | 14.2 | | 2385 | 76.7 | | 32 | 18.3 | | 60 | 15.5 | | 19 | 9.6 | | 36 | 13.4 | |
| M-H $\chi^2_1$ | 46.12 | | | 62.03 | | | 1.89 | | | 7.63 | | | 1.65 | | | 1.57 | | |
| **Sex** | | | | | | | | | | | | | | | | | | |
| Male | 410 | 12.6 | .01 | 2366 | 72.7 | .90 | 31 | 17.9 | .29 | 50 | 13.7 | .34 | 20 | 7.8 | .88 | 33 | 12.6 | .65 |
| Female | 204 | 10.3 | | 1443 | 72.6 | | 11 | 12.5 | | 31 | 11.1 | | 9 | 8.3 | | 18 | 10.9 | |
| Pearson $\chi^2_1$ | 6.56 | | | 0.02 | | | | | | | | | | | | | | |
| **Dementia** | | | | | | | | | | | | | | | | | | |
| No | 375 | 9.3 | | 2794 | 69.4 | | 31 | 16.0 | 1.0 | 55 | 10.8 | .01 | 23 | 7.6 | .58 | 34 | 10.9 | .31 |
| Yes | 239 | 19.6 | < .001 | 1015 | 83.4 | < .001 | 11 | 16.4 | | 26 | 19.0 | | 6 | 9.7 | | 17 | 14.8 | |
| Pearson $\chi^2_1$ | 96.34 | | | 92.22 | | | | | | | | | | | | | | |
| **m-CCI**[b] | | | | | | | | | | | | | | | | | | |
| Mild | 117 | 8.5 | | 866 | 62.8 | | 3 | 11.5 | .08 | 22 | 10.5 | .10 | 1 | 3.6 | .06 | 9 | 13.2 | .37 |
| Moderate | 178 | 10.3 | < .001 | 1207 | 69.9 | < .001 | 9 | 11.0 | | 22 | 11.1 | | 4 | 4.1 | | 19 | 13.9 | |
| Severe | 319 | 14.9 | | 1736 | 81.2 | | 30 | 20.0 | | 37 | 15.6 | | 24 | 10.0 | | 23 | 10.3 | |
| M-H $\chi^2_1$ | 38.55 | | | 150.61 | | | 3.13 | | | 2.63 | | | 3.62 | | | 0.81 | | |

[a]Comparisons were based on Fisher's Exact test except for age and m-CCI categories.

[b]Comparisons were based on Mantel-Haenszel chi-squared test (M-H $\chi^2$) for trend.

m-CCI = Charlson's comorbidity index (modified version).

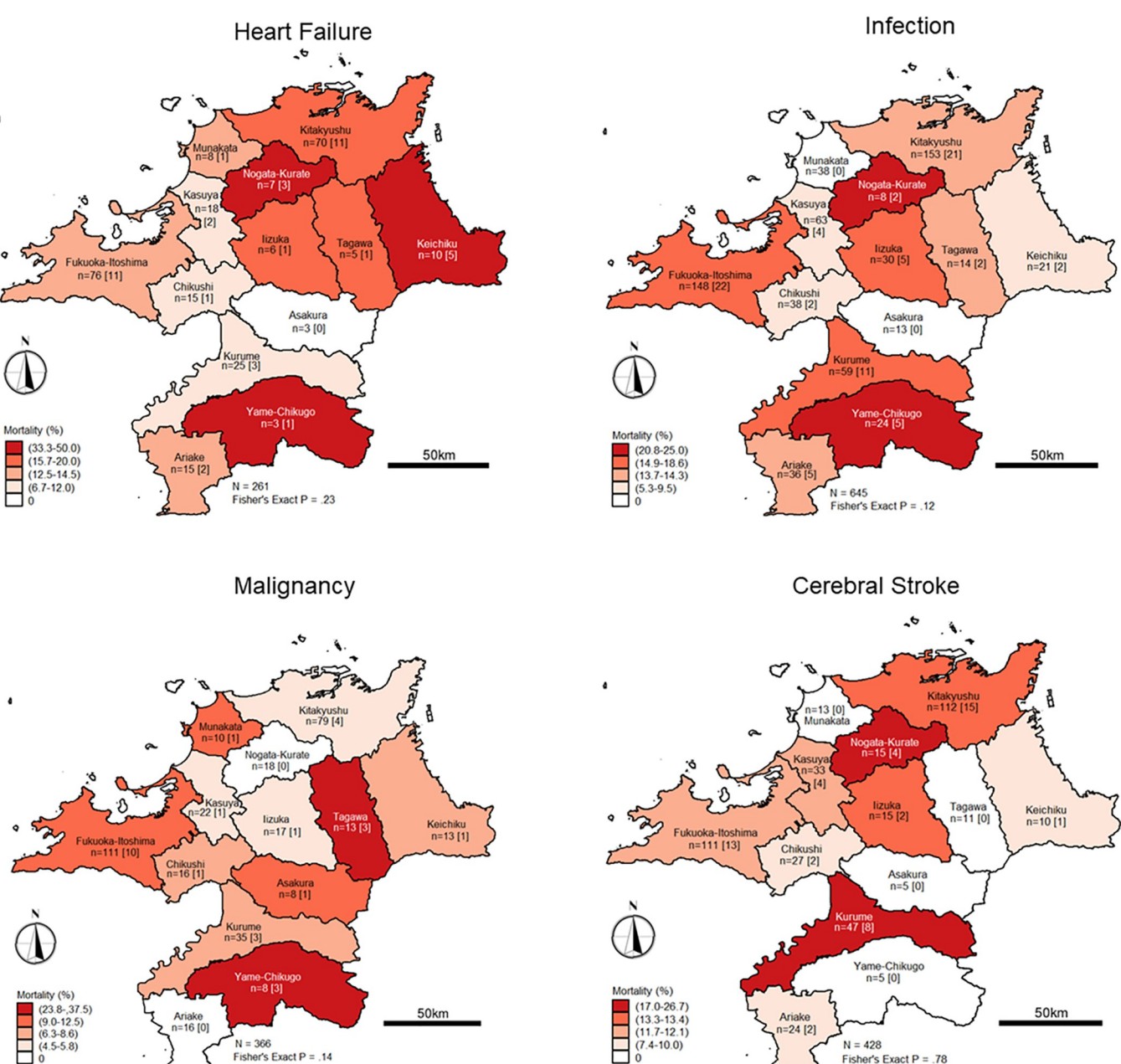

**Fig 4. Mortality cases associated with heart failure, infection, malignancy, and cerebral stroke according to SMAs (April 1, 2017–March 31, 2018).** The number of associated mortality cases is provided in square brackets. Reprinted from National Land Information Division, National Spatial Planning and Regional Policy Bureau, MLIT of Japan and maps were created using shapefiles based on Secondary Medical care Area (SMA), downloaded from the National Land Information Division Web Mapping System (Ministry of Land, Infrastructure, Transport and Tourism, Japan) at https://nlftp.mlit.go.jp/ksj/gml/datalist/KsjTmplt-A38.html, under a CC-BY license, with permission from National Land Information Division, National Spatial Planning and Regional Policy Bureau, MLIT of Japan, original copyright ©2014.

multiple logistic regression analyses were constructed by incorporating patient's variables (i.e., age, sex, dementia, *m*-CCI) in the model to control possible confounding effects. For these analyses, cases reported in SMA *Fukuoka-Itoshima* were used as reference.

The results obtained from univariate logistic regression and multiple logistic regression analyses were mostly identical. Focusing on the results of multivariable logistic regression analyses, no

**Table 3. Results of univariate logistic regression predicting all-cause mortality, hospitalization, and diagnosis-specific mortality according to secondary medical care areas.**

| | All-cause mortality | | | Hospitalization | | | Diagnosis-specific mortality[a] | | | | | | | | | | | |
| | | | | | | | Heart Failure | | | Infection | | | Malignancy | | | Cerebral Stroke | | |
| SMA | OR | 95% CI | p | OR | 95% CI | p | OR | 95% CI | p | OR | 95% CI | p | OR | 95% CI | p | OR | 95% CI | p |
|---|---|---|---|---|---|---|---|---|---|---|---|---|---|---|---|---|---|---|
| *Fukuoka-Itoshima* | Ref. | | | Ref. | | | Ref. | | | Ref. | | | Ref. | | | Ref. | | |
| *Kasuya* | 0.81 | 0.55–1.19 | .28 | 0.90 | 0.70–1.16 | .40 | 0.86 | 0.20–3.75 | .84 | 0.43 | 0.15–1.22 | .11 | 0.67 | 0.11–3.97 | .66 | 1.11 | 0.35–3.49 | .85 |
| *Munakata* | 0.80 | 0.48–1.31 | .37 | 0.99 | 0.72–1.39 | .99 | 1.14 | 0.18–7.33 | .89 | 0.07 | 0.00–1.23 | .07 | 1.53 | 0.24–9.57 | .65 | 0.27 | 0.02–4.81 | .37 |
| *Chikushi* | 0.70 | 0.48–1.04 | .07 | 0.81 | 0.64–1.04 | .10 | 0.59 | 0.10–3.54 | .56 | 0.39 | 0.10–1.50 | .17 | 0.94 | 0.15–5.62 | .94 | 0.72 | 0.17–2.95 | .64 |
| *Asakura* | 0.55 | 0.25–1.20 | .13 | 1.23 | 0.77–1.95 | .39 | 0.81 | 0.04–16.8 | .89 | 0.21 | 0.01–3.63 | .29 | 1.93 | 0.30–12.48 | .49 | 0.66 | 0.03–12.7 | .79 |
| *Kurume* | 1.06 | 0.79–1.44 | .69 | 1.28 | 1.02–1.60 | .04 | 0.89 | 0.24–3.21 | .85 | 1.33 | 0.61–2.92 | .47 | 1.04 | 0.29–3.72 | .95 | 1.57 | 0.62–4.00 | .34 |
| *Yame-Chikugo* | 1.02 | 0.61–1.72 | .93 | 0.91 | 0.63–1.31 | .60 | 3.42 | 0.41–28.4 | .26 | 1.59 | 0.56–4.52 | .39 | 6.15 | 1.39–27.11 | .02 | 0.66 | 0.03–12.7 | .79 |
| *Ariake* | 0.86 | 0.58–1.29 | .47 | 1.83 | 1.34–2.49 | < .001 | 1.05 | 0.24–4.67 | .94 | 0.99 | 0.36–2.70 | .98 | 0.29 | 0.02–5.24 | .40 | 0.81 | 0.19–3.37 | .77 |
| *Iizuka* | 0.72 | 0.42–1.23 | .23 | 1.71 | 1.16–2.51 | .01 | 1.55 | 0.23–10.5 | .65 | 1.21 | 0.44–3.38 | .71 | 0.88 | 0.15–5.25 | .89 | 1.35 | 0.31–5.84 | .69 |
| *Nogata-Kurate* | 1.03 | 0.65–1.62 | .92 | 1.32 | 0.93–1.87 | .12 | 4.43 | 0.96–20.5 | .06 | 2.16 | 0.47–9.95 | .32 | 0.26 | 0.01–4.65 | .36 | 2.86 | 0.84–9.76 | .09 |
| *Tagawa* | 0.72 | 0.43–1.22 | .22 | 0.76 | 0.55–1.04 | .09 | 1.90 | 0.27–13.4 | .52 | 1.12 | 0.27–4.70 | .87 | 3.22 | 0.82–12.63 | .09 | 0.32 | 0.02–5.70 | .44 |
| *Kitakyushu* | 0.90 | 0.71–1.14 | .40 | 2.06 | 1.72–2.48 | < .001 | 1.10 | 0.45–2.68 | .83 | 0.91 | 0.48–1.73 | .78 | 0.58 | 0.18–1.81 | .35 | 1.16 | 0.53–2.53 | .71 |
| *Keichiku* | 1.07 | 0.68–1.67 | .78 | 1.33 | 0.94–1.88 | .11 | 5.70 | 1.50–21.7 | .01 | 0.72 | 0.18–2.90 | .65 | 1.16 | 0.19–7.08 | .87 | 1.15 | 0.19–7.06 | .88 |
| Nagelkerke $r^2$ | .00 | | | .03 | | | .08 | | | .06 | | | .09 | | | .04 | | |

[a]Estimates were based on penalized maximum likelihood logistic regression with Firth correction.

Abbreviation: SMA = Secondary Medical Care Area.

significant increase in all-cause mortality risk was observed in all SMAs. Five SMAs were, however, significantly associated with increased hospitalization cases, namely *Kitakyushu* (OR 2.12, 95% CI 1.75–2.56), *Ariake* (OR 2.10, 95% CI 1.53–2.88), *Iizuka* (OR 1.90, 95% CI 1.28–2.88), *Keichiku* (OR 1.48, 95% CI 1.04–2.12), and *Kurume* (OR 1.37, 95% CI 1.09–1.74).

Analyses of diagnosis-specific mortality were also performed. However, preliminary analyses revealed that our data had the characteristics of outcome imbalance, resulting in quasi and complete data separation. Therefore, analyses were carried out using penalized maximum likelihood estimation based on Firth's correction. The use of this approach (also called Firth logistic regression) provides unbiased estimates of logistic regression, especially when dealing with issues such as rare events, quasi- and complete separation, and outcome imbalanced [35]. Its use has been validated across scientific literature [35–39].

For heart failure mortality, a significant increase in risk was only observed among patients receiving care in facilities located in *Keichiku*. Patients with heart failure were 7 times likely to die when compared to patients receiving care in facilities located in *Fukuoka-Itoshima* (OR 7.27, 95% CI 1.79–29.49). Similarly, an increase in mortality risk was observed among patients with malignancy and receiving care in facilities located in *Yame-Chikugo*. Compared with patients who were treated in facilities located in *Fukuoka-Itoshima*, we observed a statistically significant fivefold increase in mortality risk (OR 5.80, 95% CI 1.28–26.35). **Table 3** presents the results of the univariate logistic regression analyses on all-cause mortality, hospitalization cases, and diagnosis-specific mortality, and **Table 4** presents the results of the multiple logistic regression analysis for all-cause mortality, hospitalization status, and diagnosis-specific mortality.

### Analysis of LOS, total hospitalization cost, and cost per day

Data regarding LOS, total hospitalization costs, and costs per day—from 3,809 patients who had received one or more medical care requiring hospitalization—were analyzed in

**Table 4. Results of multiple logistic regression analyses on all-cause mortality, hospitalization, and diagnosis-specific mortality according to age, sex, dementia status, and secondary medical care areas (SMAs).**

| | All-cause Mortality | | | Hospitalization | | | Diagnosis-specific mortality[a] | | | | | | | | | | | |
| | | | | | | | Heart Failure | | | Infection | | | Malignancy | | | Cerebral Stroke | | |
| | OR | 95% CI | *p* | OR | 95% CI | *p* | OR | 95% CI | *p* | OR | 95% CI | *p* | OR | 95% CI | *p* | OR | 95% CI | *p* |
|---|---|---|---|---|---|---|---|---|---|---|---|---|---|---|---|---|---|---|
| **Age** | | | | | | | | | | | | | | | | | | |
| *65–74* | Ref. | | | Ref | | | Ref | | | Ref. | | | Ref. | | | Ref. | | |
| ≥75 | 1.61 | 1.33–1.96 | < .001 | 1.43 | 1.25–1.63 | < .001 | 2.14 | 0.95–4.83 | .07 | 1.93 | 1.11–3.33 | .02 | 1.82 | 0.77–4.27 | .17 | 1.35 | 0.70–2.59 | .37 |
| **Sex** | | | | | | | | | | | | | | | | | | |
| Male | Ref. | | | Ref | | | Ref | | | Ref. | | | Ref. | | | Ref. | | |
| Female | 0.76 | 0.63–0.91 | < .001 | 1.02 | 0.89–1.16 | .80 | 0.53 | 0.24–1.17 | .12 | 0.75 | 0.46–1.23 | .25 | 1.37 | 0.60–3.13 | .46 | 0.76 | 0.41–1.41 | .39 |
| **Dementia** | | | | | | | | | | | | | | | | | | |
| No | Ref. | | | Ref | | | Ref | | | Ref | | | Ref. | | | Ref | | |
| Yes | 2.09 | 1.74–2.05 | < .001 | 1.92 | 1.62–2.29 | < .001 | 1.00 | 0.65–2.23 | .98 | 1.44 | 0.83–2.51 | .19 | 0.88 | 0.32–2.40 | .78 | 1.36 | 0.71–2.59 | .36 |
| ***m*-CCI** | | | | | | | | | | | | | | | | | | |
| Mild | Ref. | | | Ref | | | Ref. | | | Ref | | | Ref | | | Ref. | | |
| Moderate | 1.12 | 0.87–1.43 | .38 | 1.32 | 1.13–1.54 | < .001 | 0.61 | 0.16–2.36 | .47 | 0.95 | 0.50–1.80 | .88 | 0.79 | 0.12–5.40 | .81 | 0.98 | 0.42–2.28 | .96 |
| Severe | 1.67 | 1.33–2.10 | < .001 | 2.59 | 2.20–3.04 | < .001 | 1.46 | 0.43–5.00 | .54 | 1.37 | 0.77–2.43 | .29 | 2.14 | 0.37–12.28 | .40 | 0.70 | 0.31–1.58 | .39 |
| **SMA** | | | | | | | | | | | | | | | | | | |
| *Fukuoka-Itosh* | Ref. | | | Ref. | | | Ref. | | | Ref. | | | Ref. | | | Ref. | | |
| *Kasuya* | 0.80 | 0.54–1.18 | .26 | 0.91 | 0.70–1.19 | .50 | 0.64 | 0.15–2.86 | .56 | 0.45 | 0.16–1.32 | .15 | 0.65 | 0.11–3.85 | .64 | 1.11 | 0.35–3.52 | .85 |
| *Munakata* | 0.80 | 0.48–1.32 | .38 | 1.00 | 0.71–1.41 | .98 | 1.52 | 0.22–10.6 | .67 | 0.06 | 0.00–1.08 | .06 | 1.77 | 0.28–11.35 | .55 | 0.31 | 0.02–5.49 | .42 |
| *Chikushi* | 0.71 | 0.48–1.06 | .09 | 0.85 | 0.66–1.09 | .20 | 0.69 | 0.11–4.24 | .69 | 0.37 | 0.09–1.43 | .15 | 0.96 | 0.16–5.84 | .97 | 0.68 | 0.16–2.83 | .60 |
| *Asakura* | 0.61 | 0.28–1.35 | .22 | 1.53 | 0.95–2.54 | .08 | 0.41 | 0.02–8.71 | .57 | 0.24 | 0.01–4.18 | .32 | 2.29 | 0.33–16.03 | .40 | 0.65 | 0.03–12.8 | .78 |
| *Kurume* | 1.07 | 0.79–1.46 | .65 | 1.37 | 1.09–1.74 | .01 | 1.04 | 0.28–3.87 | .96 | 1.24 | 0.56–2.75 | .60 | 0.98 | 0.27–3.53 | .98 | 1.49 | 0.58–3.80 | .40 |
| *Yame-Chiku* | 0.92 | 0.54–1.55 | .74 | 0.85 | 0.58–1.24 | .39 | 4.68 | 0.54–40.9 | .16 | 1.37 | 0.46–4.02 | .57 | 5.80 | 1.28–26.35 | .02 | 0.71 | 0.04–13.8 | .82 |
| *Ariake* | 0.90 | 0.60–1.35 | .62 | 2.10 | 1.53–2.88 | < .001 | 1.06 | 0.23–4.81 | .94 | 0.88 | 0.31–2.45 | .80 | 0.30 | 0.02–5.52 | .42 | 0.72 | 0.17–2.99 | .65 |
| *Iizuka* | 0.76 | 0.44–1.31 | .33 | 1.90 | 1.28–2.88 | < .001 | 1.32 | 0.18–9.44 | .79 | 1.20 | 0.43–3.37 | .73 | 1.40 | 0.22–9.03 | .73 | 1.42 | 0.33–6.16 | .64 |
| *Nogata-Kur* | 0.99 | 0.63–1.59 | .99 | 1.31 | 0.92–1.88 | .14 | 4.10 | 0.86–19.9 | .08 | 1.69 | 0.35–8.17 | .51 | 0.34 | 0.02–6.14 | .47 | 2.94 | 0.85–10.2 | .09 |
| *Tagawa* | 0.74 | 0.43–1.26 | .26 | 0.80 | 0.58–1.13 | .22 | 1.45 | 0.20–10.6 | 71 | 1.07 | 0.25–4.65 | .93 | 4.01 | 0.98–16.37 | .05 | 0.28 | 0.02–5.12 | .39 |
| *Kitakyushu* | 0.87 | 0.69–1.11 | .27 | 2.12 | 1.75–2.56 | < .001 | 1.23 | 0.50–3.05 | .66 | 0.92 | 0.48–1.75 | .81 | 0.65 | 0.20–2.06 | .46 | 1.12 | 0.51–2.46 | .77 |
| *Keichiku* | 1.15 | 0.73–1.82 | .55 | 1.48 | 1.04–2.12 | .03 | 7.27 | 1.79–29.5 | .01 | 0.67 | 0.16–2.74 | .57 | 1.25 | 0.20–7.83 | .81 | 1.12 | 0.18–6.94 | .90 |
| Nagelkerke $R^2$ | | .06 | | | .11 | | | .15 | | | .10 | | | .13 | | | .06 | |

[a]Estimates were based on penalized maximum likelihood logistic regression with Firth correction.

Abbreviation: OR = Odds Ratio; CI = Confidence Interval, m-CCI = Modified version of Charlson's Comorbidity Index, SMA = Secondary Medical Care Area.

Generalized Linear Models (GLMs). For the LOS, the result indicated significant associations in two STMs. Interestingly, patients treated in *Yame-Chikugo* and *Tagawa* were associated with a 20-day shorter LOS (β = −20.04, CI −34.61 to −5.48, *p* = .01) and an 18-day shorter LOS (β = −18.30, CI −32.14 to −4.45, *p* = .01), respectively, when compared to the patients treated in facilities located in *Fukuoka-Itoshima*.

A similar approach was used to analyze the total hospitalization costs. We did not observe any statistically significant findings. The subsequent analysis performed on costs per day identified a significant increase in costs for patients treated in facilities located in *Iizuka* ($_e$β = 1.20, CI 1.04–1.40, *p* = .02). Patients who were treated in facilities located in *Nogata-Kurate*, however, were subject to a lower cost per day compared to patients treated in facilities located in *Fukuoka-Itoshima* ($_e$β = 0.82, CI 0.70–0.95, *p* =. 01).

To examine the variability in LOS, total hospitalization cost, and cost per day, according to SMAs, Welch's one-way ANOVA was performed. The *F*-statistics indicate that LOS ($F_{12,739.18}$

**Table 5. Results of analyses of LOS, total hospitalization costs, and costs per day according to SMAs.**

| SMA | LOS | | | Total cost (US $) | | | Cost per day (US $) | | |
|---|---|---|---|---|---|---|---|---|---|
| | *M* | *SE* | β | *M* | *SE* | ₑβ | *M* | *SE* | eβ |
| *Fukuoka-Ito* | 71.67 | 2.93 | Ref. | 29,654.4 | 962.79 | Ref. | 639.14 | 17.31 | Ref. |
| *Kasuya* | 75.54 | 6.48 | 3.86 | 31,076.32 | 2116.83 | 1.0 | 630.91 | 35.85 | 1.0 |
| *Munakata* | 62.24 | 7.07 | −9.43 | 26,642.40 | 2402.14 | 0.9 | 654.50 | 49.23 | 1.0 |
| *Chikushi* | 76.20 | 6.42 | 4.52 | 32,540.84 | 2177.19 | 1.1 | 700.11 | 39.08 | 1.1 |
| *Asakura* | 63.10 | 9.65 | -8.57 | 25,730.98 | 3122.56 | 0.9 | 649.38 | 65.75 | 1.0 |
| *Kurume* | 64.96 | 4.31 | −6.71 | 27,628.56 | 1453.80 | 0.9 | 646.49 | 28.38 | 1.0 |
| *Yame-Chikugo* | 51.63 | 6.84 | −20.04** | 23,998.94 | 2522.19 | 0.8 | 572.65 | 50.21 | 0.9 |
| *Ariake* | 66.25 | 55.56 | −5.42 | 28,041.37 | 1867.93 | 0.9 | 590.73 | 32.83 | 0.9 |
| *Iizuka* | 82.50 | 9.08 | 10.82 | 33,295.77 | 2910.33 | 1.1 | 769.63 | 56.12 | 1.2* |
| *Nogata-Kurate* | 81.40 | 8.83 | 9.71 | 30,852.84 | 2657.14 | 1.0 | 523.62 | 37.62 | 0.8** |
| *Tagawa* | 53.38 | 6.43 | −18.30** | 24,420.46 | 2336.53 | 0.8 | 608.87 | 48.60 | 1.0 |
| *Kitakyushu* | 72.93 | 2.93 | 1.25 | 31,240.73 | 996.86 | 1.1 | 680.25 | 18.11 | 1.1 |
| *Keichiku* | 58.38 | 6.31 | −13.29 | 27,898.16 | 2393.95 | 0.9 | 720.17 | 51.56 | 1.1 |
| Cameron & Windmeijer's $r^2$ | | .006 | | | .005 | | | .013 | |
| Welch's ANOVA $F_{12,739.18}$ | | 2.25 | | $F_{12,737.69}$ | 1.85 | | $F_{12,764.20}$ | 3.55 | |
| | | $\omega^2$ | .002 | | | .002 | | | .004 |
| | | $p$ | .008 | | | .038 | | | < .001 |

*p < .05

**p < .01

***p < .001.

Abbreviation: LOS = length of (hospital) stay, M = Estimated Marginal Mean; SE = Standard Error, β = Beta coefficient; ₑβ = exponentiated beta coefficient; SMA = Secondary Medical Care Area.

= 2.25, $\omega^2$ = 0.002, $p$ = .01), total hospitalization costs ($F_{12,737.69}$ = 1.82, $\omega^2$ = .002, $p$ = .04), and costs per day ($F_{12,764.20}$ = 3.55, $\omega^2$ = .004, $p$ < .001), varied significantly. However, post-hoc *Games & Howell*'s pairwise multiple comparisons for unequal variance analyses showed that variations within and between SMAs were only significant when the costs per day were statistically compared. In particular, the costs per day for patients receiving care in facilities located in *Nogata-Kurate* vs. *Fukuoka-Itoshima* ($d$ = .21, $p$ = .013), in *Nogata-Kurate* vs. *Iizuka* ($d$ = .44, $p$ = .02), in *Kitakyushu* vs *Nogata-Kurate* ($d$ = .29, $p$ < .001), and Nogata-Kurate vs. Chikushi ($d$ = .34, $p$ = .02) varied statistically. **Table 5** summarizes the results of analyses using GLM on LOS, total hospitalization cost, and costs per day according to SMAs. The statistics also include the results of Welch's one-way ANOVA analyses, examining the variations in LOS, total hospitalization cost, and cost per day.

## Discussions

This study recorded a slightly higher proportion of all-cause mortality among hemodialysis patients (11.7%) compared to the national crude mortality statistics (10%) provided by the Japanese Society for Dialysis Therapy (JSDT) [1]. This result is anticipated because the analyzed claims data largely belong to older populations. As such, the data of patients aged younger than 65 years old were not included. Until December 2018, the total number of ESRD patients aged below 65 years old in Japan was 105,227—about 32% of the total ESRD patient population. Additionally, the mortality statistics provided by JSDT also considered patients receiving dialysis treatment by other modalities such as hemodiafiltration (HDF) and peritoneal dialysis

(PD), so these modalities offer different survival outcomes, as reported in several studies [40, 41]. According to a recent statistic, 37% of ERSD patients in Japan were on hemodiafiltration, and 2.8% received peritoneal dialysis regularly [1]. Nonetheless, the high number of mortality cases observed in older-age categories is consistent with the annual statistics collected and presented by the JSDT. The proportions of patients who died due to heart failure, infectious diseases, malignancy, or cerebral stroke were slightly lesser than the national statistics. However, the pattern is similar to that of reported causes of death among dialysis patients in Japan, with heart failure having the highest proportion, followed by infectious diseases and malignancy.

We used the claims-based definition of death to ascertain mortality cases. As the insurance claims data were collected primarily for reimbursement purposes—rather than for research activities—we acknowledge the risks of misclassification and coding errors. Several attempts to validate the definitions of death using insurance claims data have been documented. In one particular study, which used the health insurance claims and enrollment data of three private insurance beneficiaries between the year 2005 and 2009, the 'dead' status on the claims record had a high specificity (99.9%) and a positive predictive value (95.6%) [22]. A more recent analysis, which used claims data from the year 2012 to 2015, found that the specificity and positive predictive value of such a status were equally high for both inpatient and outpatient claims data (between 95.7% and 100%) [23]. Given such high specificity and predictive values, we believe that there is a low probability of overestimating death cases or misclassifying a deceased person as being alive due to the absence of a death record in the claims. Although both studies indicated the low sensitivity of the claims-based, and both recommended its use in combination with other determinants (e.g., disease status, comorbidity index), we still believe such estimates are improving over time, as insurance providers have been regularly conducting quality improvement audits in recent years.

A substantial number of hemodialysis patients have had one or more hospitalization episodes. This statistic makes sense, given the susceptibility of hemodialysis patients to various illnesses because of their compromised immunity status and increased frailty due to aging [42–44]. Nonetheless, our observations of both the hospitalization and mortality data revealed a trend whereby SMAs with a high number of reported mortality cases would also report many hospitalization cases i.e., *Ariake*, *Keichiku*, and *Kurume*. This observation may be consistent with previous studies that have reported increases in mortality rates following multiple admissions [45, 46]. On the contrary, such a trend was not seen in *Kitakyushu* and *Fukuoka-Itoshima*, despite a high number of hospitalization cases reported in these large, and urbanized SMAs. Perhaps these SMAs have better health care provision, including emergency care services. Furthermore, the high physician-to-population ratio in these SMAs, as reported by Maeda et al. [47], might also explain these findings. Our analysis did not show sizable variations in LOS or total hospitalization cost across SMAs. A slight increase in cost per day, however, was observed in *Iizuka*. Despite the absence of a statistical increase in LOS or total hospitalization cost, *Iizuka* had the highest marginal mean for total hospitalization costs and the longest LOS.

The findings presented in this study are also significant for clinical practice, as they provide an understanding of the magnitude of the burden of CKD and ESRD in the community. High prevalence of ESRD, associated mortality, and hospitalized patients in a community signal the need for clinicians to assume broader roles in measures against chronic kidney disease and call for a direct involvement in community health promotion programs to raise public awareness. Although this study provides limited evidence of statistical variations in mortality cases across SMAs, the results of multivariable logistic regressions suggest that patients receiving HD care in facilities located in *Kurume*, *Ariake*, *Iizuka* and *Kitakyushu* had an increase hospitalization risk. This implies that an effective strategy to improve clinical outcomes is needed. In

particular, improvement to the current level of medical care provided by facilities located in *Kurume*, *Ariake*, *Iizuka* and *Kitakyushu*, is necessary. In this study, we did not identify the reasons behind each hospitalization episode. However, we believe that a substantial number of the conditions being treated are related to CKD, such as diabetes. Since effective disease management is needed, a strong collaboration between dialysis care providers and medical institutions specializing in related diseases is necessary.

CKD itself should be widely recognized as a serious life-threatening disease, with a substantial impact on public health. Progression to ESRD will necessarily cause a patient to depend on long-term, expensive dialysis therapy. Although the treatment for ESRD in Japan is mostly provided by medical institutions specializing in renal disease, mild CKD symptoms, when presented at general medical facilities, are generally addressed with measures such as blood pressure and glucose control, and advice to reduce salt intake. Once the condition progresses to an advanced stage, specialized medical treatment has to be initiated. Even against this background, an effective medical care system that can detect and diagnose CKD early and initiate and continue appropriate treatment at an early stage is currently not in place, despite its importance. To strengthen and improve the provision of regional health care delivery, primary care physicians are recommended to collaborate with medical institutions specialized in renal disease through referral/reverse-referral and two-doctor systems. Such systems have also been proposed during the first "Kidney Disease Control Commission Meeting," held in December 2017 [11]. For this collaboration to succeed, nonetheless, the criteria for a primary care physician to refer patients to specialized medical institutions must be established and recognized.

The Ministry of Health, Labor and Welfare (MHLW) has recently recognized a shortage and uneven distribution of local medical professionals involved in CKD treatment [48]. This maldistribution in the workforce seems to originate from the existing practice of projecting the workforce that is based on the provider-to-population ratio rather than population demand or population needs [49]. Due to the rapidly changing Japanese population and disease patterns, a few researchers have deemed such conventional methods for projecting the workforce supply as no longer feasible, instead recommending the use of projection based on utilization [49–51]. While solving this issue requires thorough long-term planning at the ministerial level, a short-term solution, perhaps, could be used by developing existing human resources. Therefore, we recommend related academic societies to train health professionals, including primary care physicians, nurses, registered dieticians, and pharmacists to become certified kidney disease educators.

There are several limitations to the scope and methodology of this study. The use of insurance claims data limited the patient selection to those over 65 years old, and patients residing in one specific region in Japan. Therefore, the results of this study can only be fully generalized to older residents in Fukuoka Prefecture. Nevertheless, the analyses of claims data provide better population-based estimates and thus wider population coverage. Past studies using a similar database reported a high penetration rate (98.6%) of insurance claims data in Fukuoka prefecture [52–54]. Primary diagnosis, used as a 'surrogate' for disease-specific mortality, might not represent the actual causes of death among hemodialysis patients. Therefore, any significant relationships observed in this study should not be interpreted as 'causal,' and the results of this study must be interpreted with caution. Statistically, the analyses might provide some evidence of associations between the variables of interest. However, the regression diagnostics indicate that the logistic models might insufficiently explain the heterogeneity within the dataset. Based on the Nagelkerke $R$-squared values, the predictor variables, i.e., sex, age, dementia status, $m$-CCI categories, and SMAs, explained $\leq 15\%$ of the variations in outcome. Likewise, the calculated $r$-squared values for GLMs showed a similar pattern. The calculated effect sizes for Welch's ANOVA ($\omega^2$), on the other hand, also showed a rather small variation

in LOS, total hospitalization cost, and cost per day, despite being statistically significant. Therefore, other unmeasured factors might explain the remaining variation in outcomes. Researchers are encouraged to use a different analytical approach to better understand the magnitude and sources of these variations in the future. This study recommends future researchers to use a multilevel method and consider the inclusion of care facility variables that could potentially influence practice patterns, to better explain the heterogeneity in patient outcomes.

## Acknowledgments

We thank the Fukuoka Prefecture Wide-Area Association of Latter-Stage of Elderly Healthcare for providing the health claims data for this study.

## Author Contributions

**Conceptualization:** Aziz Jamal, Yunfei Li, Takako Fujita, Shinichiro Yoshida.

**Data curation:** Aziz Jamal, Yunfei Li, Takako Fujita, Shinichiro Yoshida, Sung A. Kim.

**Formal analysis:** Aziz Jamal.

**Investigation:** Aziz Jamal, Yunfei Li, Takako Fujita.

**Methodology:** Aziz Jamal, Akira Babazono, Shinichiro Yoshida.

**Project administration:** Akira Babazono.

**Resources:** Akira Babazono.

**Software:** Akira Babazono, Yunfei Li.

**Supervision:** Akira Babazono.

**Validation:** Akira Babazono, Takako Fujita, Shinichiro Yoshida, Sung A. Kim.

**Visualization:** Aziz Jamal, Sung A. Kim.

**Writing – original draft:** Aziz Jamal.

**Writing – review & editing:** Aziz Jamal, Takako Fujita, Sung A. Kim.

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
