## [Decision Letter · Decision Letter 0]

12 Nov 2020

PONE-D-20-25428

Elucidating small area variations in hemodialysis care and outcomes among end-stage renal disease patients in Fukuoka Prefecture, Japan.

PLOS ONE

Dear Dr. Jamal,

Thank you for submitting your manuscript to PLOS ONE. After careful consideration, we feel that it has merit but does not fully meet PLOS ONE’s publication criteria as it currently stands. Therefore, we invite you to submit a revised version of the manuscript that addresses the points raised during the review process.

We look forward to receiving your revised manuscript.

Kind regards,

Gregory Tiao, M.D.

Academic Editor

PLOS ONE

Journal Requirements:

2. We note that you have reported significance probabilities of 0 in places. Since p=0 is not strictly possible, please correct this to a more appropriate limit, eg 'p<0.0001'.

3.  We note that Figures [1 - 3] in your submission contain [map/satellite] images which may be copyrighted. All PLOS content is published under the Creative Commons Attribution License (CC BY 4.0), which means that the manuscript, images, and Supporting Information files will be freely available online, and any third party is permitted to access, download, copy, distribute, and use these materials in any way, even commercially, with proper attribution. For these reasons, we cannot publish previously copyrighted maps or satellite images created using proprietary data, such as Google software (Google Maps, Street View, and Earth). For more information, see our copyright guidelines: http://journals.plos.org/plosone/s/licenses-and-copyright.

1.     You may seek permission from the original copyright holder of Figure(s) [1 - 3] to publish the content specifically under the CC BY 4.0 license.  

Additional Editor Comments (if provided):

The authors present and interesting manuscript but need to address the issues raised by the expert panel of reviews

Reviewers' comments:

Reviewer's Responses to Questions

**Comments to the Author**

1. Is the manuscript technically sound, and do the data support the conclusions?

Reviewer #1: Yes

Reviewer #2: Partly

Reviewer #3: Partly

2. Has the statistical analysis been performed appropriately and rigorously? 

Reviewer #1: Yes

Reviewer #2: Yes

Reviewer #3: Yes

3. Have the authors made all data underlying the findings in their manuscript fully available?

Reviewer #1: Yes

Reviewer #2: Yes

Reviewer #3: No

4. Is the manuscript presented in an intelligible fashion and written in standard English?

Reviewer #1: Yes

Reviewer #2: Yes

Reviewer #3: Yes

5. Review Comments to the Author

Reviewer #1: Despite the limitations associated with a retrospective analysis and the potential sources of bias and error that can occur as a result of use of claims data, this is a very nice study with a well designed and executed statistical analysis. The only change that I would make would be to tamp down the discussion a bit - given that the outcomes were only significant in one or two of the SMAs, I'm not sure that any generalisable conclusions can be validly drawn from this and I particularly would not link this with any social determinants of health such as the phenomenon of social hospitalisation without strong direct evidence to back this up. Such conjectures just lessen validity of what is otherwise a very well thought out statistical analysis and discussion. I particularly like the fact that authors directly address the limitations of the study; both in their choice of statistical methods and in the discussion section. One other thought is that the authors could also consider adding some suggestions about the future of regional healthcare for haemodialysis patients to the discussion as well.

Reviewer #2: Topic: Elucidating small area variations in hemodialysis care and outcomes among end-stage renal disease patients in Fukuoka Prefecture, Japan.

a) Your text is talking to outcomes in terms of mortality and hospitalisation for ESRD patients on haemodialysis.

b) Remove that word small area in the topic so that it remains neutral.

c) Therefore, I would suggest your topic to be altered as follows: Elucidating variations in outcomes among older end-stage renal disease patients on haemodialysis in Fukuoka Prefecture, Japan.

Abstract

a) May you please add the name of your study design in the abstract.

b) Electronic claim records data from 5,243 patients were retrospectively reviewed. How did you select these hospital records, in other words, what was your sampling strategy?

c) Keywords: Some of the key words are not correct. Remove claims data; cost; LOS ;

Introduction

a) May you please provide information on the global prevalence of ESRD so that it becomes comparable to that one of Japan.

Methods

a) May you please highlight your study design

Data source and patient selection

a) This section has included setting of the study, study population, inclusion and exclusion criteria, sample size and sampling strategy. May you please reorganize this section under the aforementioned subheadings so that it becomes clearer to the readers

b) You only considered ESRD patients from the age of 65 years for your study and the reason should be explained clearly in your inclusion and exclusion criteria.

c) You used the sample size of 5243. Give an explanation of how you arrived at this sample size under the subheading sample size.

d) This study used anonymized claims insurance data. Thus, the requirement to obtain informed consent was waived in accordance with the Ethical Guidelines for Medical and Health Research Involving Human Subjects in Japan [https://www.mhlw.go.jp/file/06-Seisakujouhou-10600000- Daijinkanboukouseikagakuka/0000080278.pdf]. This study was also approved by the Institutional Review Board of Kyushu University (Clinical Bioethics Committee of the Graduate School of Medical Sciences, Kyushu University). This is for the subheading ethical considerations.

Results

You cannot use Descriptive Statistics as a subheading for your results section.

May you please organise your results section by including subheadings like demographic data etc.

Discussion

Your discussion is well written. Well done.

Reviewer #3: This is a study of geographic variation in dialysis-related costs and outcomes within a prefecture in Japan. The authors use private insurance data on patients >65 years old who receive regular hemodialysis to identify mortality, hospitalizations, patient characteristics, and comorbidities. Outcomes of interest include all-cause mortality, cause-specific mortality, hospitalizations, hospital length of stay, and hospital costs. Exposures include the 13 secondary medical care areas (SMAs) in the Fukuoka prefecture, patient age, comorbidity score, and presence of dementia.

According to the authors, the 3 main findings from the study are: 1) there was statistically significant variation across SMAs in length of stay, total hospitalization costs, and cost per day. 2) Some specific causes of mortality (due to heart failure and malignancy) were more likely in certain SMAs. 3) Older patients, patients with dementia, and patients with more comorbidities had a higher likelihood of mortality and hospitalization.

As the authors suggest, it is important to understand how care delivery and outcomes vary geographically. While this analysis has a potential to provide important information in this regard, there are several major limitations that should be addressed. Below is a list of major and more minor comments and suggestions:

Major:

1) Since this study is about variation in care delivery and health outcomes, the introduction would benefit from additional background and context. What specifically have past studies on this topic in ESRD and in other areas of healthcare demonstrated? How have these previous findings motivated the current study? Why is variation at the level of the SMA important? What kinds of differences across SMAs might influence health costs and outcomes? This would help to focus the study design, reporting of results, and discussion.

2) Following the above comment, the study could benefit from more of a focus. Currently, there are many different outcomes assessed and it isn’t clear why certain findings were selected for emphasis. If the focus of the study is on variation in costs and outcomes, then the highlighting of findings related to age, co-morbidities and dementia, seems less important unless the authors can demonstrate how these contributed to overall variation.

3) The authors focus on statistical significance of variation across SMAs but do not address the clinical significance of these findings. I would suggest that they focus on the primary outcomes (mortality, hospitalization, hospital costs) and try to quantify variation across SMAs in a way that can help the reader to understand its clinical significance. Consider focusing on results of the multivariable models rather than univariate comparisons.

More minor suggestions:

Introduction:

- “making up approximately 4% of the annual GDP spending on health.” This statement is unclear. Is it 4% of annual GDP or 4% of the fraction of GDP that is allocated toward health?

- “statistical variations of ESRD patients across Japan can be found … in which large variation s in ESRD incidences in Kyushu and Shikoku ..” Please clarify what type of variation has been observed. Is it variation over time or are the authors referring to differences in these two geographic areas?

- The paragraph in the introduction on HD vs. HDF made me think that this was going to be a paper on dialysis modality. However, it seems like patients receiving HDF were excluded from the analysis. Is this correct? If so, I would suggest removing this from the introduction. The exclusion of HDF could be discussed in the methods and, perhaps, in limitations.

- “Japan has recognized the varied survival outcomes of ESRD patients”. Again, what variation is recognized? Who in Japan recognized varied survival?

Methods:

- Please provide some background about Fukuoka. How big is it? Where is it? What is the population density and are there any important health characteristics worth noting?

- Briefly describe any criteria applied to the selection of co-morbidities. For example, were only certain claims examined? Did a comorbidity only have to show up once to be included?

- Why was HDF excluded?

- Discuss whether there has been any effort to validate the claims-based method of identifying mortality. If there hasn’t, include this in study limitations.

- In patient selection, mention any additional exclusion criteria applied in certain analyses. For example, were any of the analyses restricted to patients who were hospitalized at least once during the study period?

- Explain why medication costs were excluded when collecting hospital expenditures?

- Were any malignancies excluded when determining cause of death? One could imagine that some of the more common malignancies such as prostate cancer and non-melanoma skin cancer may be present for a long period of time on claims but may not be the cause of death. I suggest excluding some of these more “chronic” malignancies.

- Clarify whether assignment of patients to a prefecture was based on where they lived or where the hospitals and offices that they visit were located?

Results:

- I don’t understand this sentence: “For patients with malignancy, statistically significant difference was only observed when the m-CCI categories were statistically compared”.

- The finding that heart failure and malignancy deaths were much higher in certain areas was interesting, but made me wonder what differences there might be across prefectures. Is it possible that certain prefectures have more specialized hospitals where sicker patients with certain conditions would go to receive care? IF this is the case, would these patients be assigned to the area where they receive care, or would they be assigned to the area where they live?

- Why was the analysis of total hospital costs limited to patients who were hospitalized? This restriction seems unnecessary.

- The authors discuss multiple statistical tests. A brief description of why these tests were chosen would be informative.

Discussion:

- The authors discuss the need to limit social admissions. While I do not disagree with this position, I cannot find anything in this study that informs this discussion and would suggest removing it.

6. PLOS authors have the option to publish the peer review history of their article (what does this mean?). If published, this will include your full peer review and any attached files.

Reviewer #1: No

Reviewer #2: **Yes: **Dr Geldine Chironda

Reviewer #3: No

---

## [Author Response · Author response to Decision Letter 0]

30 Jan 2021

Academic Editor’s Comments:

1. Please ensure that your manuscript meets PLOS ONE's style requirements, including those for file naming. The PLOS ONE style templates can be found at: 

Author response:

Thank you for your comment. The revised manuscript has been carefully checked to ensure that it satisfies PLOS ONE’s style requirements.

2. We note that you have reported significance probabilities of 0 in places. Since p=0 is not strictly possible, please correct this to a more appropriate limit, e.g., 'p<0.0001'.

Author response:

Thank you for your comment. The p-values stated in both the manuscript and the tables have been corrected.

3. We note that Figures [1 - 3] in your submission contain [map/satellite] images which may be copyrighted. All PLOS content is published under the Creative Commons Attribution License (CC BY 4.0), which means that the manuscript, images, and Supporting Information files will be freely available online, and any third party is permitted to access, download, copy, distribute, and use these materials in any way, even commercially, with proper attribution. For these reasons, we cannot publish previously copyrighted maps or satellite images created using proprietary data, such as Google software (Google Maps, Street View, and Earth). For more information, see our copyright guidelines: http://journals.plos.org/plosone/s/licenses-and-copyright.

Author response:

Thank you for informing us of the requirements. We are glad to inform you that we have already received permission to use the shapefiles (used to create the maps) from the copyright holder. We have uploaded the completed Content Permission Form in the submission system.

The copyright statement has also been included in the figure caption of the copyrighted figures.

Author response:

Thank you for informing us about the requirement regarding data availability. Unfortunately, the Ethics Committee of Kyushu University has restricted data sharing. Therefore, we have changed our data disclosure statement to the following:

“The data used in our study cannot be publicly shared due to the restriction imposed by the Ethics Committee of Kyushu University. However, researchers who meet the criteria for access to this confidential data may request data access by emailing the administrative office of Bioethics Section (Medical sciences), Academic Research Cooperation Division of Kyushu University at ijkseimei@jimu.kyushu-u.ac.jp” 

5. The authors present and interesting manuscript but need to address the issues raised by the expert panel of reviews.

Author response:

Thank you for the comment. The manuscript has been extensively revised to accommodate the suggestions provided by the reviewers. We have prepared point-by-point responses to the issues raised by the expert panel of reviewers.

Reviewers' comments

Reviewer #1

Despite the limitations associated with a retrospective analysis and the potential sources of bias and error that can occur as a result of use of claims data, this is a very nice study with a well-designed and executed statistical analysis. The only change that I would make would be to tamp down the discussion a bit - given that the outcomes were only significant in one or two of the SMAs, I'm not sure that any generalisable conclusions can be validly drawn from this and I particularly would not link this with any social determinants of health such as the phenomenon of social hospitalisation without strong direct evidence to back this up. Such conjectures just lessen validity of what is otherwise a very well thought out statistical analysis and discussion. I particularly like the fact that authors directly address the limitations of the study; both in their choice of statistical methods and in the discussion section. One other thought is that the authors could also consider adding some suggestions about the future of regional healthcare for haemodialysis patients to the discussion as well.

Author response: 

Thank you very much for your kind feedback. The Discussion section has been extensively revised. We removed statements that imply the association between sociodemographic variables (i.e., sex, age, m-cci categories, dementia) and our measured outcomes, as we also feel our statistical analyses did not strongly support our previous statements. Additionally, we removed the statements regarding social hospitalization from the Discussion section. In the revised manuscript, we also added paragraphs containing our suggestions for regional healthcare to improve hemodialysis patient outcomes.

Reviewer #2

1. Topic: Elucidating small area variations in hemodialysis care and outcomes among end-stage renal disease patients in Fukuoka Prefecture, Japan.

a. Your text is talking to outcomes in terms of mortality and hospitalisation for ESRD patients on 

 haemodialysis.

b. Remove that word small area in the topic so that it remains neutral.

c. Therefore, I would suggest your topic to be altered as follows: Elucidating variations in outcomes among older end-stage renal disease patients on haemodialysis in Fukuoka Prefecture, Japan.

Author response:

Thank you for the suggestions. We have revised the manuscript title as per the recommendation. The manuscript has now been re-titled to “Elucidating variations in outcomes among older end-stage renal disease patients on hemodialysis in Fukuoka Prefecture, Japan.”

2. Abstract

a. May you please add the name of your study design in the abstract.

Author response:

The name of the study design has been added. We identified the design of the study as a retrospective cohort study.

b. Electronic claim records data from 5,243 patients were retrospectively reviewed. How did you select these hospital records, in other words, what was your sampling strategy?

Author response: 

We have added a few sentences in the Abstract detailing our sampling strategy. The sampling strategy is revised in the Abstract as follows: 

“Using an electronic claims database, we identified patients with chronic kidney disease (CKD) who had received HD care from April 1, 2017 to March 31, 2018. The CKD status was identified using International Classification of Disease, 10th revision code, and HD maintenance status was ascertained using specific insurance procedure codes. A total of 5,243 patients met our inclusion criteria and their records were subsequently reviewed. About 73% (n = 3,809) of patients had admission records during the period studied. Thus, the data regarding hospital length of stay (LOS) and admission costs were analyzed separately”. 

(Abstract, pp. 2, line 29—35)

c. Keywords: Some of the key words are not correct. Remove claims data; cost; LOS

Author response: 

Thank you for pinpointing these mistakes. Unrelated keywords have been removed.

3. Introduction

May you please provide information on the global prevalence of ESRD so that it becomes comparable to that one of Japan.

Author response: 

Thank you for the suggestion. New sentences describing the global statistics of chronic kidney disease and end-stage renal disease have been added to the Introduction section. These sentences are as follows: 

“Additionally, a report published by the Global Burden of Diseases, Injuries, and Risk Factors Study 2017 estimated that total of 697.5 million chronic kidney (CKD) cases being reported worldwide, with an estimated global prevalence as 9.5 % of the population in 2017. The global prevalence for ESRD, on the other hand, was only 0.041% [4]. Given that the world population has reached 7.60 billion in 2017, this statistic translates to a 3.1 million ESRD patients dependent on dialysis care.” 

(Introduction, pp. 3, line 52–57)

4. Methods

a. May you please highlight your study design

b. Data source and patient selection

Author response:

Thank you for your suggestion. We have added the name of the study design in the manuscript. We have also provided a detailed description of the data source and have also elaborated the patient selection process in the manuscript. 

(Methods-Data source pp. 7 –8, line 173–180; Methodology-Study design & patient identification, pp. 8, line 181–190)

c. This section has included setting of the study, study population, inclusion and exclusion criteria, sample size and sampling strategy. May you please reorganize this section under the aforementioned subheadings so that it becomes clearer to the readers.

Author response:

Thank you for your recommendation. The Methods section has been organized accordingly. We used the following sub-headings; Study location, Data source, Study design & Patient Identification, Inclusion & Exclusion Criteria; Final sample, Ethical consideration; Definition of variables, Definition of outcomes, and Statistical analyses.

(Methods, pp. 6–13, line 143–343)

d. You only considered ESRD patients from the age of 65 years for your study and the reason should be explained clearly in your inclusion and exclusion criteria.

Author response: 

In this study, we used a database that contains claims data of patients who enrolled in Latter-stage elderly health care insurance. This insurance scheme covers patients aged ≥ 65 years old. We described the database in the Methods section, under the Data source subheading. We also mentioned the reasons for the exclusion of those patients aged ≤ 65 years old in the Methods section, under the Inclusion & Exclusion criteria subheading. The reason is written as follows:

“We also excluded patients who were younger than 65 years old by April 1, 2017 (n = 224). As the insurance scheme is intended to provide coverage for those over 65 years old, we assumed the inclusion of younger patients in the database represents a coding error.” 

(See Methods-Inclusion & Exclusion criteria, pp. 8, line 196–198)

e. You used the sample size of 5243. Give an explanation of how you arrived at this sample size under the subheading sample size.

Author response:

The explanation has been provided in the Methods section under the Final sample subheading. 

(See Methods-Final sample, pp.9, line 202–209)

f. This study used anonymized claims insurance data. Thus, the requirement to obtain informed consent was waived in accordance with the Ethical Guidelines for Medical and Health Research Involving Human Subjects in Japan [https://www.mhlw.go.jp/file/06-Seisakujouhou-10600000- Daijinkanboukouseikagakuka/0000080278.pdf]. This study was also approved by the Institutional Review Board of Kyushu University (Clinical Bioethics Committee of the Graduate School of Medical Sciences, Kyushu University). This is for the subheading ethical considerations.

Author response: 

Thank you for your comment. The above sentences have been placed appropriately in the Methods section under the Ethical consideration subsection. (See Methods-Ethical consideration, pp.9, line 215 –221)

5. Results

You cannot use Descriptive Statistics as a subheading for your results section.

May you please organise your results section by including subheadings like demographic data etc.

Author response:

Thank you for the comments. The results section has been organized accordingly. We used the following subheadings to describe our findings: Patient characteristics, All-cause mortality and Hospitalization cases, Diagnosis-specific mortality, Outcomes association with SMA, Analysis of LOS, Total hospitalization costs, and Costs per day. (See Results, pp. 15–24) 

6. Discussion

Your discussion is well written. Well done.

Author response:

Thank you very much. 

Reviewer #3

1) (Major) Since this study is about variation in care delivery and health outcomes, the introduction would benefit from additional background and context. What specifically have past studies on this topic in ESRD and in other areas of healthcare demonstrated? How have these previous findings motivated the current study? Why is variation at the level of the SMA important? What kinds of differences across SMAs might influence health costs and outcomes? This would help to focus the study design, reporting of results, and discussion.

Author response:

Thank you for the feedback. We have revised the Introduction section accordingly. We have also included detailed backgrounds and the study context, along with a brief discussion of findings from past studies. We have also added a paragraph explaining the sources of healthcare variations in general and related them to the context of hemodialysis care services. We have also emphasized on the relevance and significance of this study in improving regional healthcare provision.

(See Introduction, pp. 3–6)

2) (Major) Following the above comment, the study could benefit from more of a focus. Currently, there are many different outcomes assessed and it isn't clear why certain findings were selected for emphasis. If the focus of the study is on variation in costs and outcomes, then the highlighting of findings related to age, co-morbidities, and dementia, seems less important unless the authors can demonstrate how these contributed to overall variation.

Author response:

Thank you for your comment. In the revised manuscript, the discussion on the statistical findings regarding age, sex, co-morbidities, and dementia was limited to the results obtained from the descriptive analyses. The discussion of the statistical findings of these covariates with measured outcomes (obtained from logistic regression analyses) was therefore omitted. We have also removed a few sentences describing the association of these covariates with measured outcomes in the abstract and the discussion section. 

3) (Major) The authors focus on statistical significance of variation across SMAs but do not address the clinical significance of these findings. I would suggest that they focus on the primary outcomes (mortality, hospitalization, hospital costs) and try to quantify variation across SMAs in a way that can help the reader to understand its clinical significance. Consider focusing on results of the multivariable models rather than univariate comparisons.

Author response:

Thank you for your comment. The Discussion section has been extensively revised. We also refocused the discussion on the primary measured outcomes. A paragraph was also added detailing the significance of the findings to clinical practice.

The discussion of the findings has also been carefully revised; only important results obtained from the multivariable analyses/models were discussed. 

4) (Introduction—minor suggestion) “making up approximately 4% of the annual GDP spending on health.” This statement is unclear. Is it 4% of annual GDP or 4% of the fraction of GDP that is allocated toward health?

Author response:

Sorry for the inconvenience. We have revised the statement as follows:

“The number of ERSD patients has now reached 339,841, equal to a total annual cost of ¥1.6 trillion (USD 14.9 billion). This indicates the medical cost for ESRD alone represents approximately 4% of the total health care budget of Japan.” 

(See Introduction, pp. 4, line 72—74) 

5) (Introduction—minor suggestion) “statistical variations of ESRD patients across Japan can be found … in which large variation s in ESRD incidences in Kyushu and Shikoku ..” Please clarify what type of variation has been observed. Is it variation over time or are the authors referring to differences in these two geographic areas?

Author response:

Thank you for your comment. The observed variations cited from Usami et al. (2000, 2002, 2003) refer to the annual incidence of ESRD within 11 regions in Japan over a period of 17 years. We clarified the findings of these cited works in the revised manuscript, as follows:

“Early attempts to examine the statistical variations in ESRD patients across Japan can be found in Usami et al works [6–8]. Comparing the annual number of patients with ESRD who had initiated dialysis maintenance from 1982 to 1998 across 11 regions in Japan, a significantly higher annual ESRD incidence was observed in the Kyushu, Shikoku, and Okinawa regions. The slope of regression lines presented by the authors also indicated an increasing rate of ESRD prevalence in the Kyushu, Sapporo, and Okinawa regions over a period of 17 years [6–8].” 

(See Introduction, pp. 4, line 82–88)

6) (Introduction—minor suggestion) The paragraph in the introduction of HD vs. HDF made me think that this was going to be a paper on dialysis modality. However, it seems like patients receiving HDF were excluded from the analysis. Is this correct? If so, I would suggest removing this from the introduction. The exclusion of HDF could be discussed in the methods and, perhaps, in limitations.

Author response:

This study excluded patients receiving HDF. We stated the reason for this exclusion in the Methods section, as follows:

“Hemodiafiltration (HDF) is considered a distinct treatment modality. Therefore, we did not include HDF patients in our study.”

(See Methods—Inclusion & exclusion criteria, pp.8, line 195–196)

In the Discussion section, we cited two studies that examined the survival outcomes of patients with HD and HDF. The results of these studies suggested that HDF provides better clinical and survival outcomes. Therefore, we believe the inclusion of HDF patients might underestimate the current study findings. Most studies we have found also considered HDF as a distinct dialysis modality, and HDF patients were usually excluded when the studies focused on evaluating HD patients. Therefore, we have reasons to believe that the exclusion of HDF is not a ‘limiter’ to our analyses. 

7) (Introduction—minor suggestion) “Japan has recognized the varied survival outcomes of ESRD patients”. Again, what variation is recognized? Who in Japan recognized varied survival?

Author response:

We have expanded the statements to provide more clarity. In the revised manuscript, we elaborated on the statement variations as follows:

“The Japanese government has recognized the variations in terms of ESRD prevalence, disease pattern, and patient survival outcomes at the national and prefectural level [6–10]. The annual collection of JSDT data, for example, provide vital statistical comparisons of the ESRD population and the outcomes between and among prefectures.”

(See Introduction, pp. 4–5, line 96–99) 

8) (Methods—minor suggestion) Please provide some background about Fukuoka. How big is it? Where is it? What is the population density and are there any important health characteristics worth noting?

Author response:

Thank you for your suggestion. We have added some information about Fukuoka prefecture in the Methods section under the Study Location subheading. The information includes geographical characteristics and socio-demographical characteristics. We have also included information about the organization of healthcare and the delivery of hemodialysis care services.

(See Methods—Study Location, pp.6–7, line 144–162)

9) (Methods—minor suggestion) Briefly describe any criteria applied to the selection of co-morbidities. For example, were only certain claims examined? Did a comorbidity only have to show up once to be included? Why was HDF excluded?

Author response:

We have added a brief description regarding the identification of co-morbidities from the claims record. This description is as follows:

“Both inpatient and outpatient claims records were reviewed and used to ascertain the diagnoses. For a specific diagnosis or comorbidity code to be considered valid and included in analysis, additional claims record specifying treatments, procedures, or prescriptions must present and indicative of specific conditions being treated. We did not, however, set a specific duration or a frequency for the condition must appear on the claims record as part of the inclusion criteria.” 

(See Methods—Definition of outcomes, pp.11, line 261—266)

We excluded patients who had undergone hemodiafiltration (HDF), as we consider HDF as a distinct treatment modality that offers different (or better) survival outcomes. 

10) (Methods—minor suggestion) Discuss whether there has been any effort to validate the claims-based method of identifying mortality. If there hasn’t, include this in study limitations.

Author response:

We have found two studies, which were conducted in Japan, that validate the claims-based methods for identifying mortality cases. We also added a short statement regarding the method in the Methods section. A brief discussion regarding the validity of this method was also added in the Discussion section. These statements are as follows:

“The insurance claim code 202 was used to determine mortality status. In the insurance data, such a code signifies loss of insurance eligibility due to death. Several epidemiological studies in Japan have used a claims-based definition of death previously. Past studies validating such a mortality status indicate high specificity and positive predictive values, suggesting a low likelihood of outcome misclassification [22, 23].” 

(See Methods-Definition of outcomes, pp. 10, line 241-248)

“We used the claims-based definition of death to ascertain mortality cases. As the insurance claims data were collected primarily for reimbursement purposes—rather than for research activities—we acknowledge the risks of misclassification and coding errors. Several attempts to validate the definitions of death using insurance claims data have been documented. In one particular study, which used the health insurance claims and enrollment data of three private insurance beneficiaries between the year 2005 and 2009, the ‘dead’ status on the claims record had a high specificity (99.9%) and a positive predictive value (95.6%) [22]. A more recent analysis, which used claims data from the year 2012 to 2015, found that the specificity and positive predictive value of such a status were equally high for both inpatient and outpatient claims data (between 95.7% and 100%) [23]. Given such high specificity and predictive values, we believe that there is a low probability of overestimating death cases or misclassifying a deceased person as being alive due to the absence of a death record in the claims. Although both studies indicated the low sensitivity of the claims-based, and both recommended its use in combination with other determinants (e.g., disease status, comorbidity index), we still believe such estimates are improving over time, as insurance providers have been regularly conducting quality improvement audits in recent years.” 

(See Discussion, pp. 24, line 500-514)

11) (Methods—minor suggestion) In patient selection, mention any additional exclusion criteria applied in certain analyses. For example, were any of the analyses restricted to patients who were hospitalized at least once during the study period?

Author response: 

The Methods section has been revised extensively, and appropriate subheadings have been added to organize the information. The description of the inclusion and exclusion criteria has been revised as follows: 

“We included all patients who had received HD care during the study period. This inclusion was limited to patients who received HD exclusively as the main treatment for ESRD. Thus, patients who primarily received peritoneal dialysis but also needed intermittent HD care were excluded. Hemodiafiltration (HDF) is considered a distinct treatment modality. Therefore, we did not include HDF patients in our study. We also excluded patients who were younger than 65 years old by April 1, 2017 (n = 224). As the insurance scheme is intended to provide coverage for those over 65 years old, we assumed the inclusion of younger patients in the database represents a coding error. We included patient who had received at least 24 HD sessions; thus, patients who had undergone fewer than 23 sessions were excluded (n = 625), as we could not rule out whether such short-term HD was provided to address issues related to ESRD or other conditions.”

(See Methods-Inclusion & exclusion criteria, pp.8, line 191–201)

For the analyses involving all-cause mortality, hospitalization, and disease-specific mortality cases, hospitalization status was not considered as the exclusion criteria. We, however, only analyzed the LOS, and total hospitalization cost and cost per day based on the claims data of patients who had at least one hospitalization episode. We describe this process as follows:

“A total of 5,243 patients were identified and found to meet the inclusion and exclusion criteria. Subsequently, the claims data of all 5,243 patients were retrospectively reviewed and the reported all-cause mortality, diagnosis-specific mortality, and hospitalization cases were analyzed according to 13 SMAs of Fukuoka prefecture. The data identified that 73% of HD patients (n = 3,809) had one or more hospital admission episodes during the period studied. To examine these patients’ utilization of hospital admission service, separate analyses on the LOS, total hospitalization costs, and costs per day were performed according to SMAs.”

 (See Methods-Final sample, pp. 9, line 202– 201)

12) (Methods—minor suggestion) Explain why medication costs were excluded when collecting hospital expenditures?

Author response:

Thank you for pinpointing this issue. The statement was written by mistake. Inpatient medication costs are actually included in the cost calculation. We have corrected the statement, as follows:

“These total hospitalization costs included the costs of surgery, procedures, medication, and diagnostic tests provided during hospitalization [24]” 

(See Methods-Definition of outcomes, pp. 12, line 277–278)

13) (Methods—minor suggestion) Were any malignancies excluded when determining cause of death? One could imagine that some of the more common malignancies such as prostate cancer and non-melanoma skin cancer may be present for a long period of time on claims but may not be the cause of death. I suggest excluding some of these more “chronic” malignancies.

Author response:

Thank you for the suggestion. We have checked the types of malignancy and identified five patients with the following malignancy status: prostate cancer (n = 2), early-stage breast cancer (n =1), and early-stage colorectal cancer (n = 2). Therefore, we excluded these cases from our descriptive and logistic regression analyses. Following that, we have also corrected the results of the analyses in the text, table, and figures.

14) (Methods—minor suggestion) Clarify whether assignment of patients to a prefecture was based on where they lived or where the hospitals and offices that they visit were located?

Author response:

We added a few sentences to clarify the assignment of patients to a prefecture. In the revised manuscript, the assignment of patients to a specific secondary medical care area (SMA) is described as follows:

“Claims records provide information on HD facilities. Because it is possible for a patient to receive HD care from facilities located in several SMAs, we identified the SMA based on the location of the facility for which a patient most frequently receives HD care. If we were unable to determine a specific SMA due to the complexity of claims records, SMA based on the patient’s residential address would otherwise be used.”

 (See Methods—Study design & patient identification, pp. 8, line 186–190)

15) (Results—minor suggestion) I don't understand this sentence: "For patients with malignancy, a statistically significant difference was only observed when the m-CCI categories were statistically compared".

Author response:

Sorry for the inconvenience. This sentence has been replaced with “The mortality cases of patients with malignancy were statistically different when compared according to the m-CCI categories” 

(See Results—Diagnosis-specific mortality, pp. 16, line 392–393)

16) (Results—minor suggestion) The finding that heart failure and malignancy deaths were much higher in certain areas was interesting but made me wonder what differences there might be across prefectures. Is it possible that certain prefectures have more specialized hospitals where sicker patients with certain conditions would go to receive care? IF this is the case, would these patients be assigned to the area where they receive care, or would they be assigned to the area where they live?

Author response:

We agree that there is a possibility for certain SMAs in Fukuoka prefecture to have more specialized hospitals. However, in this study, we identified SMAs based on the hemodialysis facilities. In Japan, ESRD patients who require dialysis treatment would usually refer to dialysis facilities located in the area that they live. We have provided an additional explanation regarding this matter in the Methods section.

“Claims records provide information on HD facilities. Because it is possible for a patient to receive HD care from facilities located in several SMAs, we identified the SMA based on the location of the facility for which a patient most frequently receives HD care. If we were unable to determine a specific SMA due to the complexity of claims records, SMA based on the patient’s residential address would otherwise be used.”

(See Methods—Study design & patient identification, pp.8, line 186–190)

An additional explanation was also added under the Study Location subheading, as follows:

“As patients typically require frequent hemodialysis sessions (2–3 times per week), the selection of hemodialysis facility is fixed to a specific SMA located in or close to the patient’s residential area. Patients, however, are allowed to change the assigned hemodialysis facility on specific occasions. For example, this change could be due to a change in residential address, or due to medical requirements that can only be provided by specialized hospitals in other SMA.” 

(See Methods—Study location, pp. 6, line 157–162)

17) (Results—minor suggestion) Why was the analysis of total hospital costs limited to patients who were hospitalized? This restriction seems unnecessary.

Author response:

The costs calculated in the study refers to the cost of hospital admission. It is possible that some patients did not have a claims record simply because they have never been admitted to any hospital during the study period (April 1, 2017—March 31, 2018) while still continuing to receive hemodialysis care from ambulatory hemodialysis facilities. Hemodialysis patients who were never admitted to hospitals would necessarily have no claims records at all, as well as no record hospital of stay. Because of the absence of such records, the data on these patients were automatically excluded from the analysis. To provide clarity, the following statement was added to the manuscript:

“A total of 5,243 patients were identified and found to meet the inclusion and exclusion criteria. Subsequently, the claims data of all 5,243 patients were retrospectively reviewed and the reported all-cause mortality, diagnosis-specific mortality, and hospitalization cases were analyzed according to 13 SMAs of Fukuoka prefecture. The data identified that 73% of HD patients (n = 3,809) had one or more hospital admission episodes during the period studied. To examine these patients’ utilization of hospital admission service, separate analyses on the LOS, total hospitalization costs, and costs per day were performed according to SMAs.”

(See Methods—Final sample, pp. 8, line 202–210)

Also, an additional paragraph detailing the process of the statistical analyses was also added:

“We performed the statistical analyses in two stages. The first stage was focused on the descriptive and inferential analyses of all-cause mortality, diagnosis-specific mortality, and hospitalization cases reported across Fukuoka prefecture’s SMAs. These statistical analyses used the insurance claims data of 5, 243 HD patients included in the study. The second stage was aimed at examining the LOS, total hospitalization costs, and costs per day of HD patients who were hospitalized at least once during the study period. Our claims data identified a total of 3,809 patients who had received one or more medical care episodes requiring hospitalization. The remaining 1,434 patients without such hospitalization records were excluded from these analyses as we could not accurately calculate their LOS and hospitalization cost.”

(See Methods—Statistical analyses, pp. 12, line 282–290)

18) (Results—minor suggestion) The authors discuss multiple statistical tests. A brief description of why these tests were chosen would be informative.

Author response:

We have added a few brief descriptions of the tests used in the manuscript. These descriptions are as follows:

We considered age groups and m-CCI categories as ordered categorical variables. Therefore, the Mantel-Haenszel Chi-squared test to determine associated trends was used as recommended by several statisticians [25].

(See Methods—Statistical analyses, pp. 12, line 295–297)

The goodness-of-fit of the logistic models was determined via regression diagnostics using Cragg & Uhler's (Nagelkerke) pseudo-R-squared test. This test value is useful in examining the fitness and the complexity of the logistic model, as it presents a normalized version of the R-squared value computed from the likelihood ratio. Thus, the calculated pseudo-R-squared value has a connection with the Wald statistics for overall association [26, 27.]

(See Methods—Statistical analyses, pp.13, line 304–307)

This model provides superior estimates compared to a normal linear regression model due to its alternative approach in dealing with problems associated with skewed LOS and cost data distribution [28].

(See Methods—Statistical analyses, pp. 13, line 310–312)

The goodness-of-fit of GLM was assessed based on Cameron & Windmeijer’s R-squared test [30]. To the best of our knowledge, it is the only statistical test that provides information on the GLM model fitness based on calculated model variance.

(See Methods—Statistical analyses, pp. 13–14 line 322–325)

“Distribution normality was subsequently inspected using a histogram, and statistically assessed using the Anderson-Darling normality distribution test. This normality distribution test was chosen because it allows the specification of exponential, lognormal, and gamma distributions, from which the critical values are calculated [32]—thus offering a greater test sensitivity.”

(See Method—Statistical analyses, pp. 14, line 332–334)

As equal variances were not assumed, F-statistics based on Welch’s ANOVA were reported. Welch ANOVA provides unbiased estimates, and its use has been recommended by many statisticians when dealing with unequal variances in data [33, 34].

(See Methods—Statistical analyses, pp. 14, line 335–337)

Therefore, analyses were carried out using penalized maximum likelihood estimation based on Firth's correction. The use of this approach (also called Firth’s logistic regression) provides unbiased estimates of logistic regression, especially when dealing with issues such as rare events, quasi- and complete separation, and outcome imbalanced [35]. Its use has also been validated across scientific literature [35–39].

(See Results—Outcomes association with SMA, pp. 20, line 429–435)

19) (Discussion—minor suggestion) The authors discuss the need to limit social admissions. While I do not disagree with this position, I cannot find anything in this study that informs this discussion and would suggest removing it.

Author response:

Thank you for the suggestion. We have removed the statements regarding ‘social hospitalization’ from the Discussion section.

---

## [Decision Letter · Decision Letter 1]

18 Mar 2021

PONE-D-20-25428R1

Elucidating variations in outcomes among older end-stage renal disease patients on hemodialysis in Fukuoka prefecture, Japan

PLOS ONE

Dear Dr. Jamal,

Thank you for submitting your manuscript to PLOS ONE. After careful consideration, we feel that it has merit but does not fully meet PLOS ONE’s publication criteria as it currently stands. Therefore, we invite you to submit a revised version of the manuscript that addresses the points raised during the review process.

We look forward to receiving your revised manuscript.

Kind regards,

Gregory Tiao, M.D.

Academic Editor

PLOS ONE

Journal Requirements:

Additional Editor Comments (if provided):

Please address the minor corrections requested by the reviewers

Reviewers' comments:

Reviewer's Responses to Questions

**Comments to the Author**

1. If the authors have adequately addressed your comments raised in a previous round of review and you feel that this manuscript is now acceptable for publication, you may indicate that here to bypass the “Comments to the Author” section, enter your conflict of interest statement in the “Confidential to Editor” section, and submit your "Accept" recommendation.

Reviewer #2: (No Response)

Reviewer #3: (No Response)

2. Is the manuscript technically sound, and do the data support the conclusions?

Reviewer #2: Yes

Reviewer #3: Yes

3. Has the statistical analysis been performed appropriately and rigorously? 

Reviewer #2: Yes

Reviewer #3: Yes

4. Have the authors made all data underlying the findings in their manuscript fully available?

Reviewer #2: Yes

Reviewer #3: Yes

5. Is the manuscript presented in an intelligible fashion and written in standard English?

Reviewer #2: Yes

Reviewer #3: Yes

6. Review Comments to the Author

Reviewer #2: Thank you authors for addressing the comments raised. Here are the few comments to address:

1. Keywords: Retrospective is not a key word, remove. May you add the word outcome as a key word.

2. Methods section in the manuscript: Organization of the methods section is paramount for the readers to follow and understand clearly. May you please reorganize this section in the following order: study design, study location, population (Patient identification), Inclusion and exclusion criteria, sample size, data source, definition of variables, definition of outcomes, data analyses and ethical considerations.

3. Study design and patient identification should be separated.

Reviewer #3: The authors have responded to my suggestions and addressed my concerns. I only have one additional suggestion. The focus of the study is on variation in care. While the authors now dedicate part of the discussion to address the clinical significance of findings about variation, much of the discussion section focuses on crude rates of various health outcomes. Less of a focus on findings from crude rates in the discussion would help to align the discussion text with the manuscripts focus, which is on geographic variation.

7. PLOS authors have the option to publish the peer review history of their article (what does this mean?). If published, this will include your full peer review and any attached files.

Reviewer #2: **Yes: **Dr Geldine Chironda

Reviewer #3: No

---

## [Author Response · Author response to Decision Letter 1]

27 Mar 2021

Reviewer #2

Thank you authors for addressing the comments raised. Here are the few comments to address:

1. Keywords: Retrospective is not a key word, remove. May you add the word outcome as a key word.

Author response:

Thank you for the suggestion. The key word ‘retrospective’ has been removed and replaced with ‘outcome’.

2. Methods section in the manuscript: Organization of the methods section is paramount for the readers to follow and understand clearly. May you please reorganize this section in the following order: study design, study location, population (Patient identification), Inclusion and exclusion criteria, sample size, data source, definition of variables, definition of outcomes, data analyses and ethical considerations.

Author response:

Thank you for your kind guidance. We have organized the methods section accordingly. We have ordered the sub-sections as followings: Study design, Study location, Population (Patient identification), Inclusion and exclusion criteria, Sample size, Data source, Definition of variables, Definition of outcomes, Data analyses, Ethical consideration.

3. Study design and patient identification should be separated.

Author response:

Thank you for your suggestion. We have separated the content accordingly. In the revised manuscript, the explanation of study design and patient identification is provided under two separate subheadings (i.e., Study design, Population (Patient identification)). 

Reviewer #3

1. The authors have responded to my suggestions and addressed my concerns. I only have one additional suggestion. The focus of the study is on variation in care. While the authors now dedicate part of the discussion to address the clinical significance of findings about variation, much of the discussion section focuses on crude rates of various health outcomes. Less of a focus on findings from crude rates in the discussion would help to align the discussion text with the manuscripts focus, which is on geographic variation.

Author response:

We appreciate your suggestion. The paragraph has been revised accordingly. We removed the sentences mentioning the crude rates and replaced them with sentences highlighting outcome variations that were observed in several areas (i.e., Kurume, Ariake, Iizuka and Kitakyushu). These observed variations were based on the result of multivariable logistic regression analyses. (see Discussion Section, p.25, Line 528—541 )

---

## [Decision Letter · Decision Letter 2]

12 May 2021

Elucidating variations in outcomes among older end-stage renal disease patients on hemodialysis in Fukuoka Prefecture, Japan

PONE-D-20-25428R2

Dear Dr. Jamal,

We’re pleased to inform you that your manuscript has been judged scientifically suitable for publication and will be formally accepted for publication once it meets all outstanding technical requirements.

Kind regards,

Gregory Tiao, M.D.

Academic Editor

PLOS ONE

Reviewers' comments:

Reviewer's Responses to Questions

**Comments to the Author**

Reviewer #2: All comments have been addressed

2. Is the manuscript technically sound, and do the data support the conclusions?

Reviewer #2: Yes

3. Has the statistical analysis been performed appropriately and rigorously? 

Reviewer #2: Yes

4. Have the authors made all data underlying the findings in their manuscript fully available?

Reviewer #2: Yes

5. Is the manuscript presented in an intelligible fashion and written in standard English?

Reviewer #2: Yes

6. Review Comments to the Author

Reviewer #2: Thank you for addressing all the comments raised. Congratulations for producing a great manuscript.

7. PLOS authors have the option to publish the peer review history of their article (what does this mean?). If published, this will include your full peer review and any attached files.

Reviewer #2: **Yes: **Dr Geldine Chironda

---

## [Editor Report · Acceptance letter]

17 May 2021

PONE-D-20-25428R2 

Elucidating variations in outcomes among older end-stage renal disease patients on hemodialysis in Fukuoka Prefecture, Japan. 

Dear Dr. Jamal:

I'm pleased to inform you that your manuscript has been deemed suitable for publication in PLOS ONE. Congratulations! Your manuscript is now with our production department. 

Kind regards, 

on behalf of

Dr. Gregory Tiao 

Academic Editor

PLOS ONE